# Realizing reversible phase transformation of shape memory ceramics constrained in aluminum

Wangshu Zheng ®[1], Yan Shi[1,2], Lei Zhao[1], Shuangyue Jia[1], Linghai Li[1], Chee Lip Gan ®[3] ✉, Di Zhang ®[1] ✉ & Qiang Guo ®[1] ✉

Small-scale shape memory ceramics exhibit superior shape memory or superelasticity properties, while their integration into a matrix material and the subsequent attainment of their reversible tetragonal-monoclinic phase transformations remains a challenge. Here, cerium-doped zirconia (CZ) reinforced aluminum (Al) matrix composite is fabricated, and both macroscopic and microscopic mechanical tests reveal more than doubled compressive strength and energy absorbance of the composites as compared with pure Al. Full austenitization in the CZ single-crystal clusters is achieved when they are constrained by the Al matrix, and reversible martensitic transformation triggered by thermal or stress stimuli is observed in the composite micro-pillars without causing fracture in the composite. These results are interpreted by the strong geometric confinement offered by the Al matrix, the robust CZ/Al interface and the local three-dimensional particle network/force-chain configuration that effectively transfer mechanical loads, and the decent flowability of the matrix that accommodates the volume change during phase transformation.

Zirconia ($ZrO_2$)-based ceramics, as a representative of shape memory ceramics (SMCs), can undergo a reversible martensitic phase transformation between tetragonal and monoclinic structures. Depending on the testing temperatures, the martensitic transformations can induce shape memory effect (SME) or superelasticity (SE) in the ceramics[1,2], making them promising candidates for various applications in energy dissipation[3,4] and transformation toughening[5]. Since the 1980s, a lot of research efforts have been dedicated to tailoring the martensitic transformation temperature (MTT) and the corresponding critical resolved shear stress (CRSS) of shape memory zirconia via grain size regulation and element doping (e.g., Ce, Y, Mg, Ca, etc.), which endowed a reversible deformation strain up to 0.5–2% and a high recovery ratio of 95–100%[6,7]. However, these SMCs, usually in the polycrystalline form, suffer from severe intrinsic brittleness and intergranular fracture, which are associated with the considerable

volume change (4–5%) and shear strain (~ 14–15%) developed during martensitic transformation[5].

In the past decade, the brittle nature of $ZrO_2$ has been mitigated by constructing miniature specimens (e.g., micro-/nano-pillars[8], single/oligo-crystals[9–12], nano-fibers[13,14]), and structures formed thereof (e.g., honeycombs via 3D printing[15], cellular foams via freeze casting[16,17], and granular packings[3,18]), where the ceramic is relatively free to expand/contract because of the high specific surface area and/ or the removal of grain boundaries. Nevertheless, it remains a challenge to integrate these small-volume shape memory ceramics into a matrix material and realize their reversible phase transformation in a confined state without causing fracture. Overcoming this hurdle would be both scientifically and technologically crucial for the SMCs to be used in engineering applications that involve bulk structural parts and in intelligent sensors/actuators that require mechanical robustness.

[1]State Key Lab of Metal Matrix Composites, Shanghai Jiao Tong University, 800 Dongchuan Road, Shanghai 200240, China. [2]Zhejiang Academy of Special Equipment Science, 211 Kaixuan Road, Hangzhou 310020, China. [3]School of Materials Science and Engineering, Nanyang Technological University, 639798 Singapore, Singapore. ✉e-mail: clgan@ntu.edu.sg; zhangdi@sjtu.edu.cn; guoq@sjtu.edu.cn

In this work, we fabricate aluminum (Al) matrix composites reinforced by cerium-doped shape memory zirconia (CZ) single-crystals clusters using a powder metallurgy approach. The stability of the tetragonal phase in the composite, and the martensitic transformation behavior of the constrained shape memory ceramics subjected to thermal and stress stimuli are examined. Reversible martensitic transformation of the CZ single-crystals clusters under the Al matrix constraint is observed at a small scale, which is rationalized by the robust CZ/Al interface that effectively transfers mechanical loads and the decent plastic flow of the Al matrix that accommodates the volume change during martensitic transformation.

## Results

### Formation and microstructure

Figure 1a, b schematically illustrates the fabrication procedure of the as-fabricated CZ/Al composites and the sample morphology at each step of the process. Single-crystalline shape memory zirconia clusters having a composition of $Ce_{0.12}Zr_{0.88}O_2$ (mol.%) were synthesized via gel-casting methods[4] (Methods). The average particle size of CZ was measured to be approximately 480 nm, using a particle size analyzer (Omni, Brookhaven) (Supplementary Fig. 1). After ball milling for 3 hr, the Al powders were reshaped into thin flakes, and the CZ particle clusters were embedded on the surface of these Al flakes (Supplementary Fig. 2b, d). The composite obtained from subsequent densification processing steps inherited the interconnected nature of the particles in its microstructure (Fig. 1b). Such a typical network structure of the particles in two/three-dimension was further demonstrated at a large view in Supplementary Figs. 3, 4, and Fig. 1c (via high-resolution X-ray computed tomography microscopy, XRT), respectively. The volume fraction of CZ in the composite was determined to be ~15.4% by XRT, aligning well with the prescribed value from composite fabrication (~16.7%). In addition, the CZ particles were preferably located at the Al grain boundaries, as evidenced by the electron back scattering diffraction (EBSD) and energy dispersive spectroscopy (EDS) images shown in Fig. 1d and Supplementary Fig. 3b, c. The particles stabilized the Al grain boundaries during hot pressing and effectively refined the Al grain size in the as-fabricated composite (~3.1 μm), as compared to that of the corresponding pure Al (~5.1 μm) (Fig. 1e).

Figure 1f displays the weight percentages of the monoclinic and tetragonal phases in the as-synthesized CZ particles, as-milled CZ/Al composite powders, and the as-fabricated CZ/Al composite. The corresponding X-ray diffraction (XRD) pattern is shown in Supplementary Fig. 5. The austenite start and finish temperatures ($A_s$ and $A_f$) and the martensite start temperature ($M_s$) of the CZ particles were measured to be $A_s = 304$ °C, $A_f = 470$ °C, and $M_s = 181$ °C, respectively (Supplementary Fig. 1b), while the martensite finish temperature ($M_f$) was lower than room temperature and cannot be obtained. This indicates an incomplete martensitic transformation in the CZ particles, resulting in 17 wt.% tetragonal (austenite) phase residual and 83 wt.% monoclinic (martensite) phase at room temperature in the as-synthesized particles. Compared to the pristine state, the CZ particles in the as-milled CZ/Al composite powders had a slight increase in the fraction of the tetragonal phase to 19 wt.%. Interestingly, upon hot pressing and subsequent cooling to room temperature, all the monoclinic phases in the CZ/Al composite, with 100% relative density (Supplementary Table 1), were found to transform to the tetragonal phase. Moreover, no $Al_3Zr$ peak was observed in the XRD pattern (Supplementary Fig. 5), indicating that the content of this interfacial reaction product was lower than the detection limit (~1 wt.%, if any). In other words, full austenization of CZ particles with structural integrity was achieved when they were constrained in the Al matrix. Similar phenomena were also found in zirconia-reinforced steel[19] and ceramic matrix composites[20–22], but never reported in soft metal matrices with high chemical activity such as pure Al.

Figure 2a shows the typical transmission electron microscopy (TEM) image of the CZ/Al composite, where the interfacial regions (boxed in Fig. 2c) were magnified and analyzed in detail in Fig. 2c, d. As shown in Fig. 2b, a clearly defined, ~5 nm thick interfacial layer was sandwiched between the CZ particle and the Al matrix, and the corresponding fast Fourier transformation analysis (FFT, Fig. 2b inset) revealed the presence of Al and tetragonal $ZrO_2$ on each side of the interfacial layer, consistent with the XRD measurements (Fig. 1b and Supplementary Fig. 5). Moreover, the diffraction spots of Al and $ZrO_2$ were superimposed on a diffuse ring without those corresponding to the possible interfacial reaction product ($Al_3Zr$), suggesting the amorphous nature of the interfacial layer. This amorphous interfacial layer was further proved to be alumina ($Al_2O_3$), determined from energy dispersive spectroscopy (EDS) linear mapping (Supplementary Fig. 6), which may come from the oxidization of Al and may enhance the cohesion between the CZ particles and the Al matrix. In addition, there was no significant elemental inter-diffusion between the CZ particles and the Al matrix, and the interface was fairly sharp, as suggested by the EDS results shown in Fig. 2d.

### Thermal response of the bulk composites

Figure 3a displays the representative 2θ-temperature-intensity contour map for the CZ single-crystal clusters. The data were obtained by in-situ XRD scanning at 50 °C → 500 °C → 50 °C, with an interval of 50 °C between every two scans and a dwell time of 5 mins (Supplementary Fig. 1c). Clearly, the CZ particles were thermally transformed between the monoclinic phase and the tetragonal phase, and the peak at ~30° as the characteristic of the tetragonal phase became much stronger after the heating-cooling treatment, indicating a greater amount of the tetragonal phase after the thermal cycling. In stark contrast, in the case of the CZ/Al composite (Fig. 3b), the tetragonal structure was kept throughout the entire temperature range, and no monoclinic peaks were detected. Due to the thermal expansion effect, the tetragonal characteristic (101) peak was shifted from approximately 29.90° to 29.74° after heating to 500 °C and shifted back to 29.84° after cooling to 50 °C. Such a minor, permanent shift of 2θ by −0.06° after one cycle of heating/cooling indicates that the CZ tetragonal lattice was expanded, most likely resulting from the partial release of the Al matrix constraint. Such a proposal is corroborated by the shift of the Al (111) peak during the heating process in Supplementary Fig. 7a.

To further examine the thermal stability of the tetragonal phase in the CZ/Al composite, we annealed the as-synthesized CZ particles and the as-fabricated CZ/Al composite for 5 min at 500 °C, well above the austenite finish temperature ($A_f = 470$ °C, Supplementary Fig. 1b). As depicted in Fig. 3c, the CZ particles in the composite after annealing retained a pure tetragonal phase, whilst there was a significant increase in the fraction of the tetragonal phase in the freestanding CZ particles (17 wt.% to 58 wt.%), which is in line with in-situ XRD results (Fig. 3a, b). Subsequently, the four sample sets, the as-synthesized CZ, the annealed CZ, the as-fabricated CZ/Al composite, and the annealed CZ/Al composite, were cryogenically treated in liquid nitrogen (around −196 °C) for 5 min. Afterwards, notable reductions in the fraction of the tetragonal phase were observed in the as-synthesized CZ particles (from 17 wt.% to 10 wt.%) and in the annealed CZ particles (from 58 wt.% to 29 wt.%). Strikingly, the tetragonal phase was mostly stable in the CZ/Al composites, where its fraction was kept at 100 wt.% in the as-fabricated CZ/Al composite and showed a slight reduction to 94 wt.% in the annealed composite (Fig. 3c). Multiple (15) annealing (500 °C) - cryogenic (−196 °C, liquid nitrogen) treatment cycles in Fig. 3d, e further revealed the role of Al matrix constraint. Contrary to expectations, the tetragonal phase ratio basically kept unchanged (around 94 wt.%) during the following 15 thermal-cryo cycles (Fig. 3e). In addition, all of the characteristic peaks had a more broadened full-width at half-maximum (FWHM, from 0.11° to 0.26°) and were shifted from 38.42° to

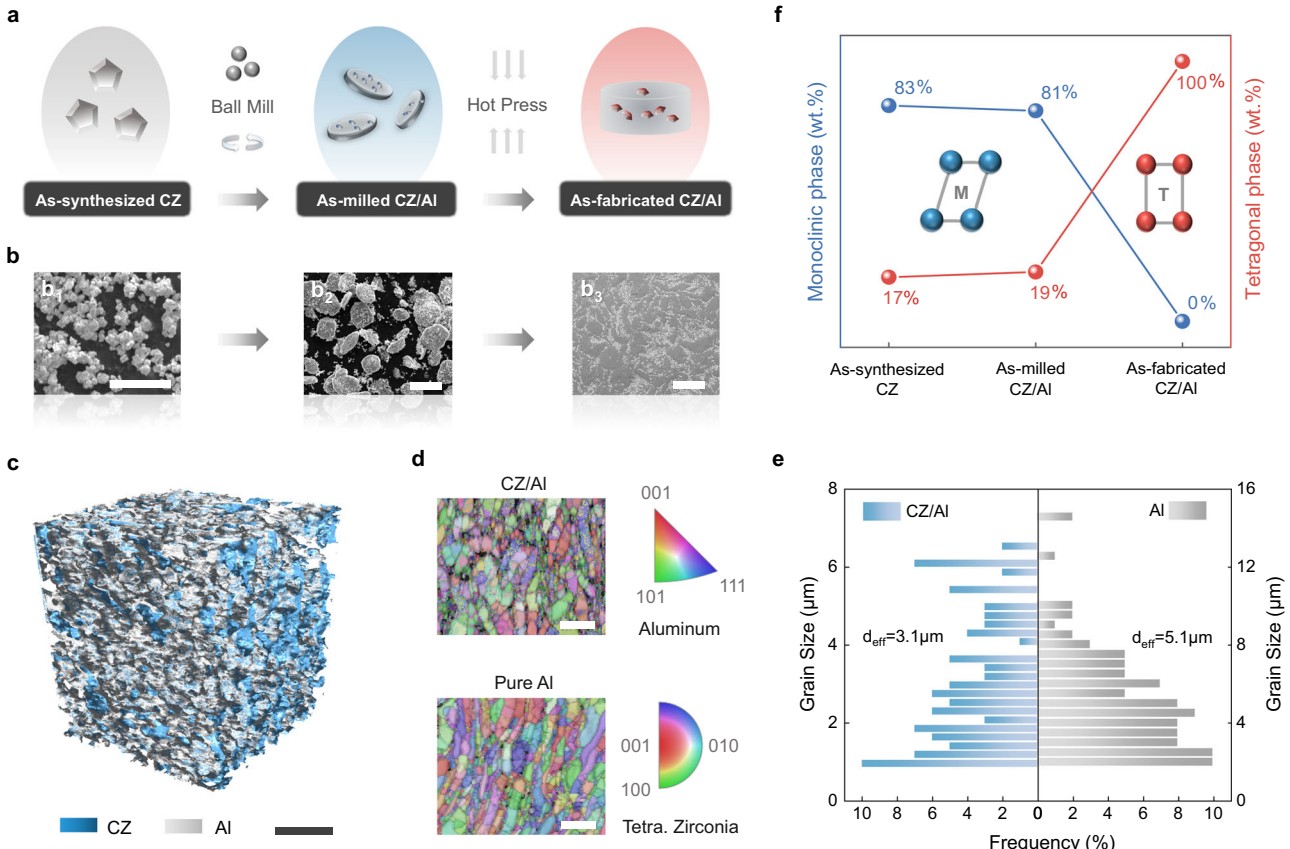

**Fig. 1 | Formation, 3-Dimensional architecture, and phase constitute of the as-fabricated CZ/Al composites. a** Schematic illustration of the as-synthesized cerium-doped shape memory zirconia (CZ) particles, the as-milled CZ/Al mixed powders, and the as-fabricated CZ/Al composite. **b** The representative scanning electron microscopy (SEM) images for each stage during fabrication are also provided in **b₁-b₃**, respectively. **c** High-resolution X-ray computed tomography microscopy (XRT) volume renderings of CZ/Al composites, where the color renderings of CZ and aluminum (Al) are blue and grey, the CZ volume fraction is estimated to be ~15.4%. **d** Electron backscattering diffraction (EBSD) of CZ/Al and pure Al, mapped with Kikuchi band contrast and crystal orientation contrast (Z axis). The average grain size ($d_{eff}$) of Al was fitted and shown in (**e**). **f** Phase constitution at room temperature (25 °C) measured by X-ray diffraction (XRD), where T and M represent tetragonal and monoclinic structure with red and blue color, respectively. Scale bar, (**b₁**) 5 μm; (**b₂-b₃**) 20 μm; (**c**) 50 μm; (**d**) 10 μm.

38.82° with increasing number of cycles (Supplementary Fig. 7c and Supplementary Table 2). According to the Williamson-Hall method[23] (Supplementary Methods), such a trend was likely to originate from the increment of dislocation density during the thermal-cryo cycles (from $3.82 \times 10^{12}$ m⁻² to $8.63 \times 10^{12}$ m⁻²), which further inhibit the martensitic transformation and stabilized the tetragonal phase (Supplementary Fig. 7d). In comparison with the martensitic transformation behavior in the freestanding CZ particles, it can be concluded that the Al matrix constraint suppressed the martensitic transformation in the CZ/Al composites and may significantly change the transformation temperatures.

**Compressive response of the bulk composites and micro-pillars**
The representative compressive stress-strain responses of the as-fabricated CZ/Al composite and the corresponding unreinforced Al matrix are manifested in Fig. 4a. The compression tests on the pure Al micro-pillars show a very similar strength (5% flow stress of $\sigma_{5\%} = 173 \pm 11$ MPa) and strain hardening behavior to that of the bulk Al samples under compression ($\sigma_{5\%} = 168 \pm 3$ MPa), except the fact that the plastic flow of micro-pillars was populated with discrete strain bursts, which were associated with dislocation-boundary interactions and commonly observed in small-scale samples[24]. Both bulk CZ/Al composite ($\sigma_{5\%} = 266 \pm 5$ MPa) and the CZ/Al micro-pillar (labeled as S1, $\sigma_{5\%} = 391$ MPa) show significant strengthening over pure Al, suggesting good strengthening capability of the CZ particles. These results are both in line with the hardness test and nano-indentation

data in Supplementary Table 1. The compressive strength of the composite micro-pillar was ~2.3 times that of the pure Al pillars. The energy absorbance of the composite pillar was also tremendously enhanced, reaching 54.0 MJ/m³, more than doubling that of pure Al pillars (24.1 MJ/m³).

In addition to the micro-pillar S1, two more representative micro-pillars (S2, S3) were fabricated from different locations of the composite, which exhibited the compressive strength of 282 MPa and 545 MPa, respectively. The stronger micro-pillars S1-S3 as compared to the corresponding bulk composite may be caused by the effectively higher CZ content in the pillar. In particular, the regions rich in CZ particle clusters (as depicted in Fig. 1c) were generally selected for micro-pillar fabrication, and the actual CZ volume fraction in pillars S1-S3 was approximately ~30 vol.%, 67 vol.% and 70 vol.%, respectively, estimated by the area fractions of CZ and Al in the TEM cross-section (Supplementary Fig. 8). They are much higher than the nominal 16 vol.% in bulk samples, calculated from the 30 wt.% weight percentage, and the densities of zirconia (6.0 g/cm³) and Al (2.7 g/cm³). The observation as such was verified by the estimation of Young's modulus of the CZ/Al micro-pillar (Supplementary Methods), where the micro-pillar having higher Young's modulus possessed higher CZ content (Supplementary Fig. 9). In addition, the spatial distribution of CZ particles in pillar S1 was in an arc-like arrangement, while in pillars S2 and S3 the particles spread over most of the cross-section areas. The typical morphologies of CZ/Al composite micro-pillars S1-S3 before and after compression are shown in Fig. 4c. The pillars S2, S3 after

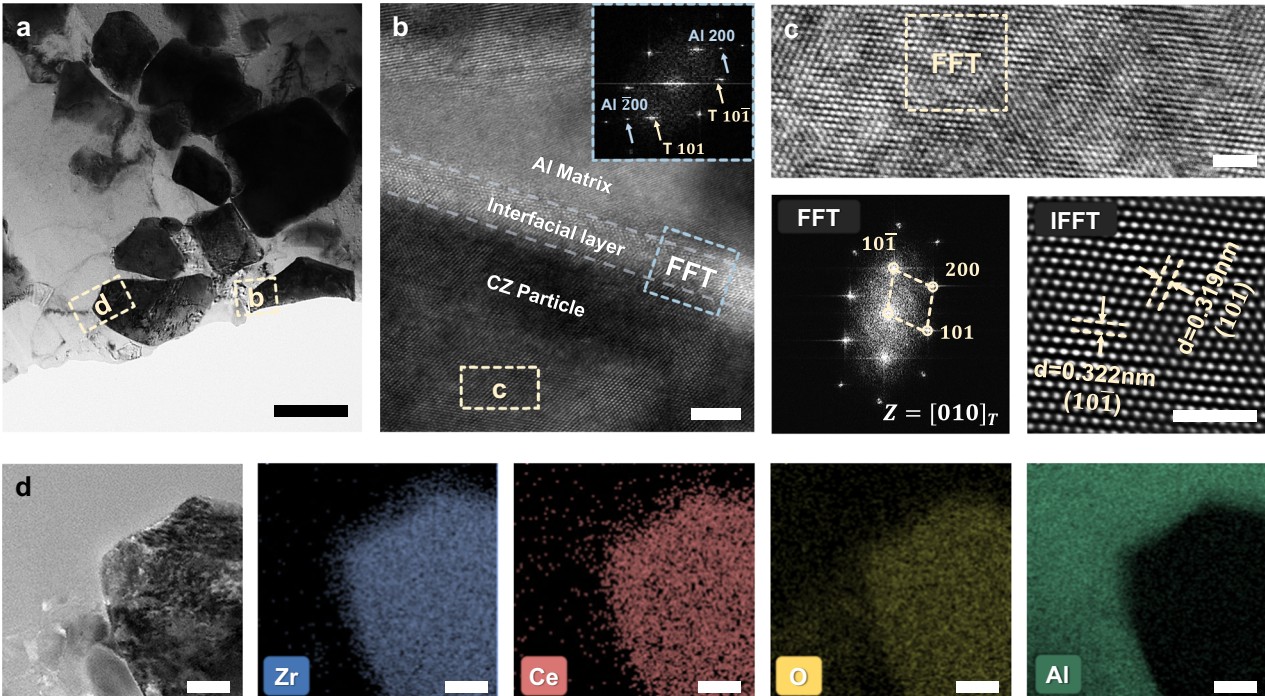

**Fig. 2 | Microstructural characterization of the as-fabricated CZ/Al composites.** **a** Bright-field (BF) transmission electron microscopy (TEM) image of the as-fabricated CZ/Al composite. **b** High-resolution TEM analysis of the interface between CZ and Al in the region boxed in (**a**); inset is the corresponding fast Fourier transformation (FFT) pattern. **c** Crystal lattice image, FFT, and corresponding inverse FFT (IFFT) patterns of CZ in the as-fabricated composite. **d** HAADF-STEM image of the as-fabricated CZ/Al composite and energy dispersive spectroscopy (EDS) color mapping of Zr (blue), Ce (red), O (yellow), and Al (green) elements in the region boxed in (**a**). Scale bar, (**a**) 500 nm; (**b**) 5 nm; (**c**) 2 nm; (**d**) 20 nm. Tetragonal-T, aluminum-Al.

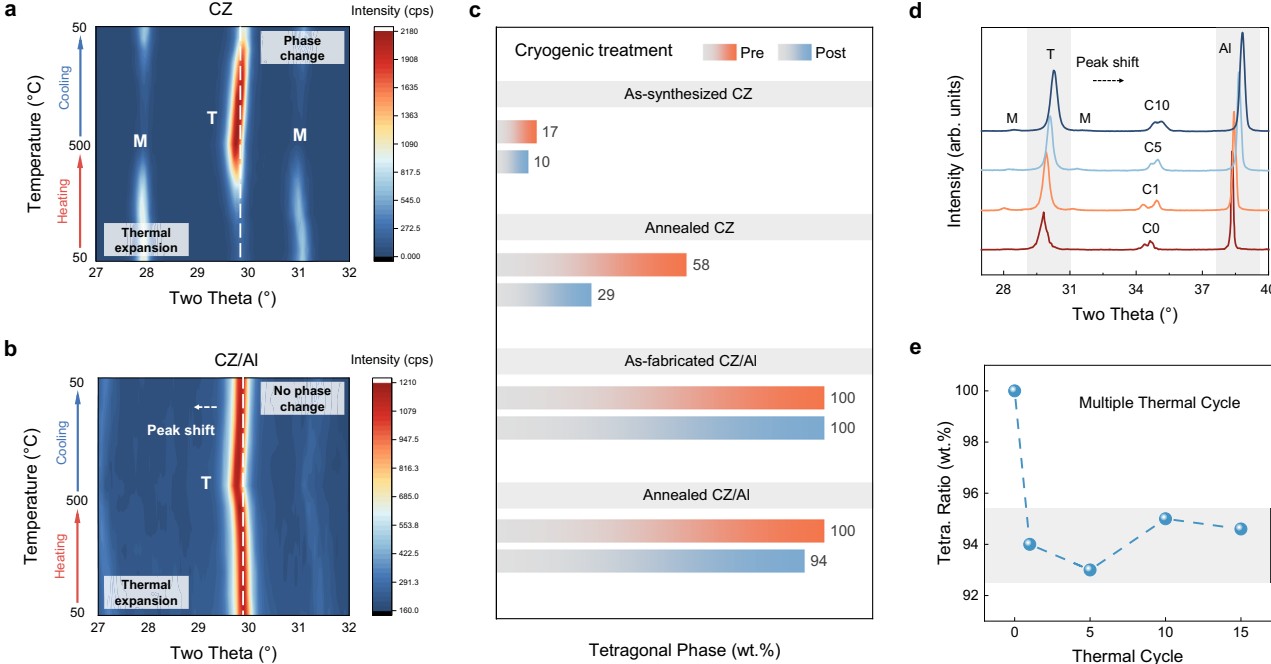

**Fig. 3 | Thermal response of the as-fabricated CZ/Al composite. a, b** Contour map of 2θ-temperature-intensity obtained from in-situ high-temperature XRD measurement of **a** CZ single-crystal particles and **b** CZ/Al composite. **c** The tetragonal phase ratio of the as-synthesized/annealed CZ, and the as-fabricated/annealed CZ/Al composite, before and after the cryogenic treatment. **d** XRD spectrums of the CZ/Al composite under multiple annealing - cryogenic treatment. The corresponding tetragonal phase ratio in **d** is shown in (**e**). Notes: C0 - as-fabricated CZ/Al; C1, C5, C10, C15 - CZ/Al composite after 1,5,10,15 cycles of thermal treatment; Tetragonal-T, monoclinic-M, aluminum-Al. Source data are provided as a Source data file.

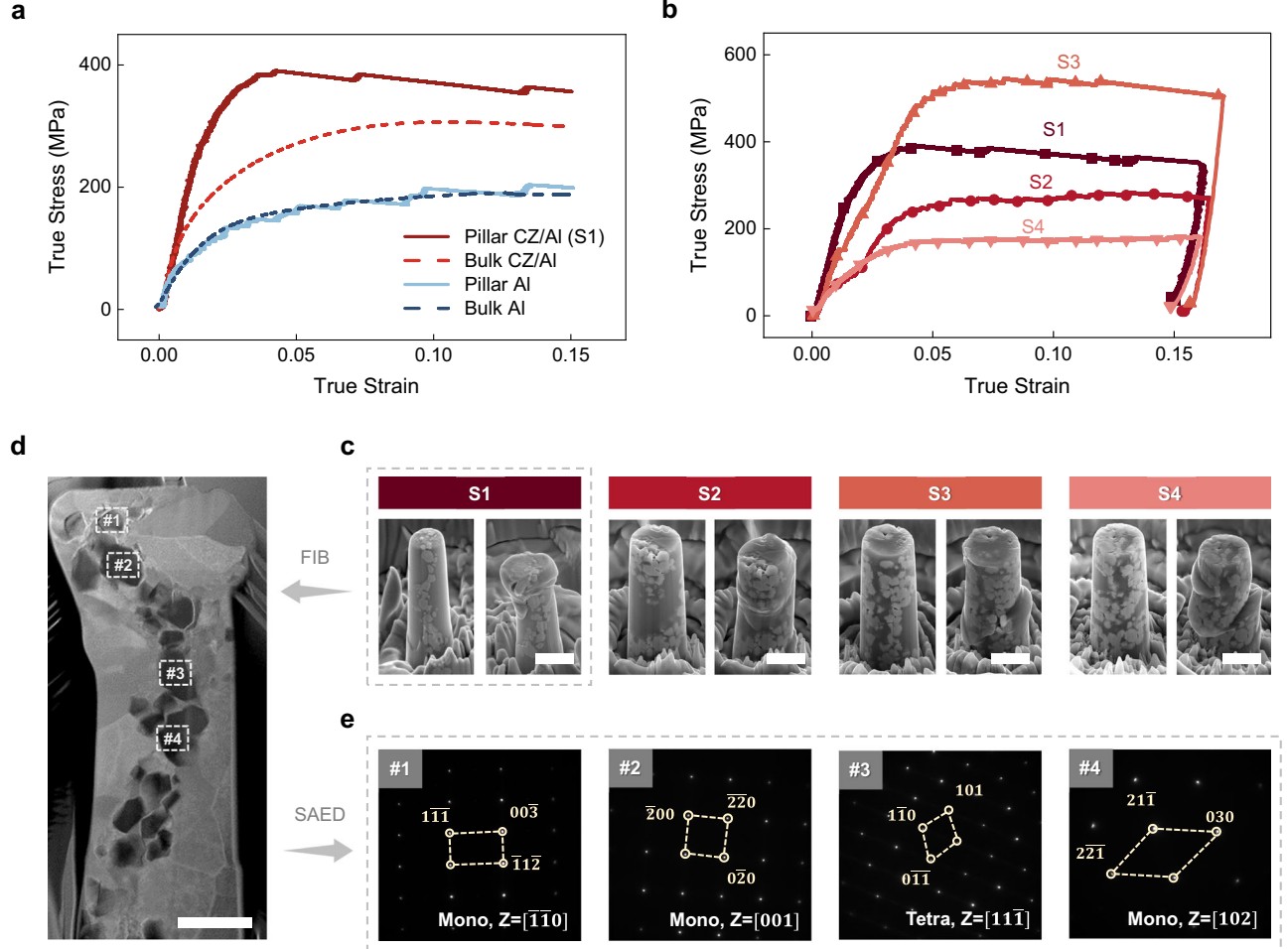

**Fig. 4 | Compressive response of the as-fabricated CZ/Al composite and post-mortem microstructural analysis. a** True stress-strain curves of CZ/Al composite and pure Al (dash line: bulk compression test; solid line: micro-pillar compression test, corresponding to the pillar S1; red: CZ/Al; blue: Al). **b** True stress-strain curves of CZ/Al micro-pillar S1, S2, S3, and S4. **c** SEM image of pre-compression (left) and post-compression (right) CZ/Al micro-pillar S1 to S4. **d** TEM cross-section prepared from the post-compression pillar S1, with selected area electron diffraction (SAED) patterns taken from #1-#4 CZ particles in (**e**). Scale bar, (**c**) 2 μm; (**d**) 1 μm.

compression experienced severe plastic deformation on their lateral surface (evidenced by the bulges due to local extrusion of the Al matrix), which were analogous to the deformation behavior of micro-pillar S1. In spite of the different particle assembly patterns in the 3 pillars, most of the CZ particles were interconnected with each other, forming clusters embedded in the Al matrix (force-chain configuration[25]). This ensured effective load transfer from the Al matrix to different particles and therefore, the mechanical response of

the 3 pillars was found to be similar. We also intentionally fabricated a representative pillar S4 with discretely distributed particle clusters (without a force-chain configuration). In stark contrast to the geometries of pillar S1-S3, three unconnected CZ particle clusters were laminated with Al grains across the gauge length of pillar S4 (Supplementary Fig. 8). Herein, a prominent shear occurred in the post-compression pillar S4 (Fig. 4c), leading to a totally different deformation mode and significantly lower compression strength (~180 MPa), even at an apparently high CZ volume ratio of ~ 55 vol.%. The CZ particle content and distribution in the pillars were summarized in Table 1.

**Table 1 | Summary of cerium-doped shape memory zirconia (CZ) particle content and distribution in the pillar, and the transformational behaviors of micro-pillars S1 to S4**

| Micro-pillar | S1 | S2 | S3 | S4 |
|---|---|---|---|---|
| CZ Distribution | Arc | Interconnected clustering | Double arc | Laminated |
| Force-chain? | Yes | Yes | Yes | No |
| CZ Content/vol.% | 30 | 67 | 70 | 55 |
| Deformation mode | Load transfer through CZ chains; local extrusion of Al | | | Al shear |
| Reversible transformation? | Yes | Yes | Yes | Yes |

The actual CZ content in each pillar was estimated from the corresponding cross-section scanning transmission electron microscopy (STEM) images. The estimated CZ volume fractions in the micro-pillars were likely to be their upper limit, as the particles' dimension in the out-of-plane direction is smaller than the pillar's dimension. (Al - aluminum).

**Stress-induced martensitic transformation in the micro-pillars**
We further examined the structure of the 15 individual CZ particles via selected area electron diffraction (SAED) in the TEM cross-section lamella of the typical deformed pillar S1 (Fig. 4d). Approximately ~60% of the CZ particles in the post-compression pillars were transformed into monoclinic phase. Such phase changes were slightly detected in the compressed bulk CZ/Al composite, as indicated by the relatively small peak at ~ 28° in the XRD pattern (Supplementary Fig. 10). Caution has to be taken that the conventional XRD measurement on bulk samples can only probe a thin depth (up to ~ 10 μm) into the specimen, and cannot provide a reliable estimate on the total amount of transformed CZ in the composite. In the following, we selected four representative CZ

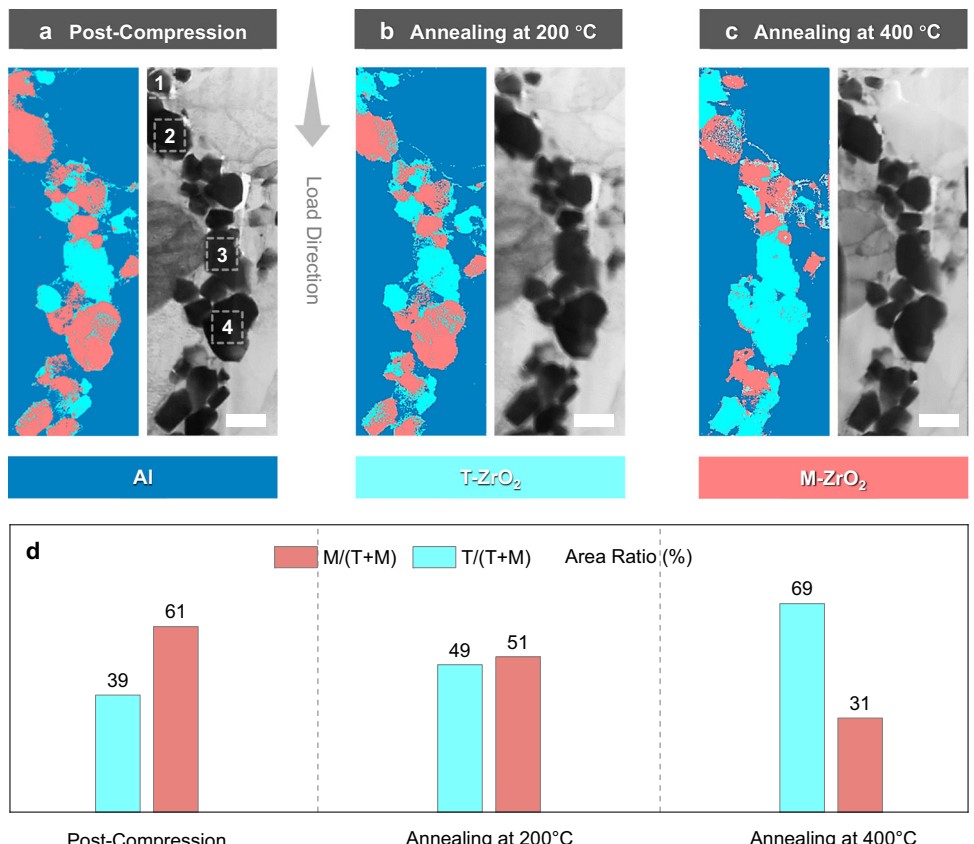

**Fig. 5 | Thermal response of the TEM cross-section lamella made from the post-compression CZ/Al composite micro-pillar S1. a–c** Precession electron diffraction (PED) bright-field image and phase map of the sample at the states of **a** post-compression (25 °C), after annealing at **b** 200 °C, and **c** 400 °C, where Al, T and M represent aluminum, tetragonal and monoclinic with dark blue, light blue, and pink color, respectively; the compression direction (from top to bottom) is indicated by the arrow. **d** Area ratio of different phases determined from the PED images in **a**–**c**. Scale bar, (**a**–**c**) 500 nm.

particles marked as #1-#4 in the micro-pillar S1 (Fig. 4d), for the discussion of reversible phase transformation. As evidenced in Fig. 4e, particles #1, #2, and #4 experienced stress-induced martensitic transformation, whilst particle #3 remained in the tetragonal phase after compression. This indicates that not all the particles can be triggered for martensitic transformation, which was likely to result from the local stress at the particle being lower than the critical stress required for the phase transformation, and/or the crystal orientation of the particle being unfavorable for the transformation. As the t→m martensitic transformation can induce ~4–5% volume expansion in the CZ particles[10] and thus produce compressive stress on the Al matrix, the Al matrix surrounding those particles going through phase transformation was observed to have abundant geometrically necessary dislocations (GNDs) near the particle/matrix interface (Supplementary Fig. 11a), while no interfacial delamination or fracture across the matrix was found. These observations imply that the ductile Al matrix can well accommodate the transformation strain of the CZ particles, and the CZ/Al interfacial bonding was strong enough to retain the coherence of the interface.

Stress-induced martensitic transformation was further studied at different stress levels within micro-pillars S2, S3, and S4. Similar to the case in micro-pillar S1, stress-induced martensitic transformation occurred in all three typical pillars (Supplementary Fig. 12). Using the total area of each phase as the benchmark for their volume fraction, the highest monoclinic phase ratio (68%) was probed in pillar S3 with the highest compressive strength (545 MPa) among all the pillars. On the contrary, only 13% monoclinic phase was detected in pillar S4

which suffered from severe Al shear, indicating a strong correlation between the stress and the transformation ratio.

**Thermally-induced reverse martensitic transformation**

The post-compression TEM cross-section lamella of micro-pillar S1 shown in Fig. 4d was annealed at 200 °C (lower than $A_s$ = 304 °C), 400 °C (between $A_s$ and $A_f$ = 470 °C), and 600 °C (above $A_f$) successively, to explore the thermally induced reverse martensitic transformation behaviors under different annealing conditions. The phase compositions of the annealed CZ particles were probed by precession electron diffraction (PED); however, examination of the 600 °C-annealed sample was not successful due to the bending of the specimen. After annealing, the morphology of the TEM cross-section lamella and the distribution of CZ particles in the Al matrix remained unchanged. Nevertheless, a considerable crystallographic evolution in the crystal structure of the CZ took place (Fig. 5a–c). The percentage of the tetragonal phase in CZ particles gradually increased with increasing annealing temperature, from ~39% in the deformed state to ~49% in the 200 °C-annealed state, and finally to ~69% in the 400 °C-annealed state (Fig. 5d). Analysis on the same #1-#4 CZ particles defined in Fig. 4d revealed that, after 200 °C annealing, the #1 particle first went through the reverse martensitic transformation and was converted back to the tetragonal phase. In comparison, the #2 and #4 particles were kept in the monoclinic phase. After 400 °C annealing, the #4 particle transformed to the tetragonal phase as well. The corresponding SAED patterns of #1-#4 particles presented in Supplementary Fig. 13 are in line with the PED results. Interestingly, after further annealing at 600 °C, #1 particle switched to the monoclinic phase again (Supplementary Fig. 13), probably as a result of the more

significant softening of Al and the subsequent relaxation on the geometrical constraint near the pillar top. The transformational behavior of the other three micro-pillars S2, S3, and S4 after 400 °C annealing agreed with that of S1 (Supplementary Fig. 14, Table 1 and Supplementary Table 3). More than half of the transformed particles (50%, 79%, and 62%, respectively) were converted back to the tetragonal phase, respectively.

## Discussion

The one-way tetragonal-to-monoclinic (t → m) phase transformation of zirconia is extensively adopted in steel- and ceramic-matrix composites, with the aim of toughening the matrix upon mechanical deformation and failure[26,27]. However, the reversible transformation between the two phases of zirconia in a confined state without causing fracture has never been realized previously. The underlying mechanism for the reversible phase transformation of shape memory zirconia particles constrained in Al at a small scale in this study is threefold. First, the tetragonal phase is stabilized in the as-fabricated CZ/Al composites at room temperature. Specifically, the substantially higher thermal expansion coefficient of Al ($-2.4 \times 10^{-5}$ /K[28]) over that of CZ ($-1 \times 10^{-5}$ /K[29]) caused compressive stress on CZ particles upon heating. When the hot-pressed CZ/Al composite was cooled down below the martensitic start temperature ($M_s = 181$ °C), the confinement of the Al matrix prevented the tetragonal CZ particles from transforming into the monoclinic phase. Such a geometric constraint successfully inhibited the t → m phase transformation (Supplementary Fig. 15a, b), exhibiting a totally different phenomenon from that of granular packings[30]. Impressively, the tetragonal phase of CZ particles was kept stable under both annealing at elevated temperatures and treatment at cryo-temperatures (Fig. 3), suggesting the strong magnitude of the geometric constraint.

Secondly, the robust CZ/Al interfaces associated with an $Al_2O_3$ interfacial layer (Fig. 2b and Supplementary Fig. 15d, h) allowed sufficient load transfer between the Al matrix and the CZ particles. This was further complemented by the network structure of CZ particles in the composite pillars, providing broad choice for designing multiple particle geometries within the micro-pillars to construct the force-chain configuration (Supplementary Fig. 15c), so that the CRSS was eventually achieved to trigger the t → m transformation in the composite pillars upon uniaxial compression. Estimated by a simple rule-of-mixture, the uniaxial stress shared on the CZ particles in the pillar S1 shown in Fig. 4 was approximately 120 MPa, using a CZ volume fraction of 30 vol.% and a 5% flow stress of 391 MPa. This is consistent with the ~100 MPa uniaxial stress needed to instigate the martensitic transformation in 12 mol.% $CeO_2$-$ZrO_2$ granular packings, where the CZ particles had a similarly good connectivity for efficient load transfer as the particle distribution in the composite pillar[3] (Fig. 4). More rigorously, considering the interconnected nature of the CZ particles in the composite pillar, the maximum local shear stress at the CZ particle-particle contact was estimated from the Hertzian contact model,

$$\tau_{\max} = 0.3 \sqrt[3]{\frac{3 F_e E^{*2}}{2\pi^3 R^2}} \tag{1}$$

where, $R = 240$ nm is the radius of curvature (corresponding to the average diameter ~480 nm of the CZ particles), and $E^* = 100$ GPa is the contact modulus[10]. $F_e = 6$ MPa is the average effective contact normal stress (estimated from uniaxial stress of 120 MPa divided by 20 particles). $\tau_{\max}$ was then calculated to be approximately 0.64 GPa. The shear stress of ~0.64 GPa falls in the range of 0.29–4.35 GPa reported for the CRSS of t → m transformation in cerium-doped zirconia[31]. In other words, the observation that the stress-induced martensitic transformation occurred in the CZ/Al composite pillar but not prominently in its bulk counterpart may be explained by the higher

content of CZ particles, and their local three-dimensional particle network/force-chain configuration in the pillars.

Finally, during iso-thermal annealing of the post-compression pillars, the softening and the ensuing flow of the Al matrix partially relaxed the geometric confinement on the CZ particles, and easily accommodated the volume contraction of the reverse martensitic transformation (as compared to the volume expansion of CZ particle upon t → m transformation), converting a large amount of the monoclinic phase back to the tetragonal phase again (Fig. 5 and Supplementary Fig. 15e, f). Indeed, the number of dislocations (GNDs) at the Al matrix were found to be notably reduced after annealing (Supplementary Fig. 11), a clear evidence of matrix softening. Even though the annealing temperature of 200 °C was markedly lower than the austenite start temperature of the free-standing CZ particles ($A_s = 304$ °C), a considerable fraction of the monoclinic phase still transformed to the tetragonal phase, indicating that the effective transformation temperatures were significantly reduced by the Al matrix constraint.

The much stronger composite as compared to the pure Al matrix was likely to be rendered by a combined effect of CZ load-sharing, a refined microstructure (Al grain size reduced from ~5.1 μm in pure Al to ~3.1 μm in the composite, shown in Fig. 1), and the strain hardening of the Al matrix derived from the GNDs near the CZ/Al interfaces in the composite (Supplementary Fig. 11). On the other hand, the high energy absorbance of the composite was imparted by the elastic and plastic deformation of the Al matrix, and more noteworthy, the stress-induced martensitic transformation of the CZ particles. Considering the high critical stress for the martensitic transformation and the large transformation strain (up to ~10%), the CZ particles can be unique and very effective reinforcement to strengthen and toughen Al alloys.

It is noteworthy that the reversible transformation as such comes at a cost of plastic (irreversible) deformation in the matrix. Herein, any application requiring cyclic loading can only be feasible well below the strength of the matrix. The future research direction may lie in the design of matrix materials with high intrinsic mechanical strength and large elastic strain[32]. Furthermore, the CZ-Al composite used as a model material in this study was characterized by a moderate microstructural inhomogeneity, where the force-chain configuration of the CZ particles was lacking in some local regions in the composite, as evidenced by the pillar S2 (Supplementary Fig. 14a) and its notably low strength than other pillars examined (Table 1). Undoubtedly, for more efficient phase transformation of the shape memory ceramics in the composite, the force-chain network of the ceramics should be uniformly populated across the entire body of the composite. This could be realized by fabricating ceramic preforms using vibration-assisted particle self-assembly[33], direct foaming[15] or sacrificial templating method[16,34,35] to create 3D interconnected particle networks, followed by metal infiltration to obtain an interpenetrating architecture of the two phases (i.e., interpenetrating composites, or IPCs[36,37]). The composite with high spatial particle connectivity is a promising candidate to realize the reversible phase transformation at the bulk level.

In conclusion, we fabricated Al matrix composite reinforced by 30 wt.% shape-memory CZ particles using a powder metallurgy approach. Full austenitization in the CZ particles was achieved when they were constrained by the pure Al matrix, and the tetragonal phase of the particles remained stable when they were annealed at elevated temperature above the austenite finish temperature, or cryogenically treated at liquid nitrogen temperature. Uniaxial compression tests on both bulk composite and composite micro-pillars revealed that their mechanical strength and energy absorbance were more than doubled over those of the pure Al matrix (in the pillar form). More encouragingly, uniaxial compression on the composite micro-pillars triggered the transformation of up to ~70% of the particles in the pillars, which breaks through the transformation limit (~40%) in the granular packing[3]. More than half of the transformed particles were converted back to the tetragonal phase upon subsequent annealing at 400 °C,

over the austenite start temperature of the particles. No fractures were observed in the vicinity of the CZ particles that have undergone martensitic transformation, and dense GNDs were found in the Al matrix close to the CZ/Al interfaces. Our study conferred the demonstration of reversible martensitic transformation of shape memory zirconia in a constrained state in densified composite micro-pillars without causing fracture. This can be attributed to the strong geometric confinement offered by the Al matrix, the robust CZ/Al interface and local three-dimensional particle network/force-chain configuration that effectively transferred mechanical loads, and the decent flowability of the matrix that accommodated the volume change during phase transformation.

As discussed previously, the stress-induced, one-way tetragonal-to-monoclinic phase transformation of shape memory ceramics provides a transformation toughening mechanism upon deformation and failure of the matrix, when the shape memory ceramic is incorporated in matrices of heavy metals (e.g., Nb[38], Ta[39], and Fe[40]) and other high-strength ceramics (such as cubic phase $ZrO_2$[21] and $Al_2O_3$[26]). From the perspective of structural applications, the transformation-induced large lattice strains of shape memory ceramics (-10% in shear strain and -5% in volume change[2]) may render the derived metal-, ceramic-, or polymer-matrix composites with high strength-toughness synergy via transformation toughening[5]. The achievement of reversible martensitic phase transformation of shape memory ceramics constrained in light metals (Al as manifested here) without causing fracture, on the other hand, may tremendously extend the application arena of these smart composites to sensors, actuators, and damping devices that require lightweight, high strength, and multi-functionality. Specifically, as compared with the conventional shape memory alloys such as NiTi[41] and Cu-Al-Ni[42], the CZ/Al composite developed in this work is superior in terms of much lower density (-3.22 g·cm$^{-3}$ in Supplementary Table 1), and wide operational range of temperature (100–500 °C), with similar actuation force/magnitude (-10$^2$ MPa). These findings suggest the possibility of realizing multiple cycles of shape memory effect or superelasticity in the CZ/Al composites for future structural and functional applications.

## Methods
### Materials
Single-crystalline shape memory zirconia particles having a composition of $Ce_{0.12}Zr_{0.88}O_2$ (mol.%, hereafter referred to as CZ) were synthesized via gel-casting methods[4]. The nitrogen-atomized pure aluminum (Al) powders were purchased from Henan Yuanyang Powder Technology Co., Ltd. (99.99% purity, $d_{50} = 10$ μm).

### Preparation of the bulk composites
30 wt.% CZ particles were co-milled with Al powders for 3 hr in a planetary ball mill (QM-3SP4, Nanda Instrument Plant, China) with a ball-to-powder ratio of 10:1 and a rotation speed of 200 rpm under the protection of high-purity argon. Then, the composite powders (as-milled CZ/Al powders) were put into a die, and the sintering was conducted at a temperature of 480 °C under high-purity argon. The holding time and pressure for sintering were set as 1 hr and 600 MPa, respectively. Finally, the composite was naturally cooled down to room temperature in the mold (as-fabricated CZ/Al composite). The 30 wt.% CZ concentration was found to be the upper limit for fully densified composite. Pure Al sample without the addition of CZ particles was also fabricated based on the above procedures (Pure Al) for comparison.

### Preparation of the micro-pillars and transmission electron microscopy (TEM) cross-sections
Focused ion beam (FIB) was used to fabricate micro-pillars (2–3 μm diameter, aspect ratio of 2:1 − 3:1) from the polished surface of the as-fabricated CZ/Al composites. Milling was conducted in three steps.

The first large crater at 30 kV and 65 nA, a designed size at 30 kV and 1.0 nA, and final reducing tapering at 5 kV and 13 pA. TEM cross-sections of the as-fabricated CZ/Al composites and micro-pillars were prepared via FIB lift-out methodology (with a thickness of 50–200 nm). After the last step for thinning the specimen, the parameters of 5 kV and 48 pA were set to remove potential Pt, Ga$^+$, and avoid irradiation-induced phase transformation in zirconia[43–45]. Each time before TEM observation, it was cleaned by an argon ion polisher (Fischione, Model 1040 NanoMill) with a milling voltage of 500 eV and milling current of 100 μA (at a tilt angle of ±10° and milling time of 30 mins each side), to remove the residual defective surface layer on the specimen, as a result of oxidation and Pt/Ga$^+$ implantation.

### Characterization techniques
X-ray computed tomography microscopy (XRT; Zeiss Xradia Versa 620, Carl Zeiss AG, Oberkochen, Germany) was performed at 140 kV, 21 W with 40X objective lens to quantitatively measure sample morphologies in three dimensions via absorption contrast tomography. The collected projection images were reconstructed via Dragonfly software. The two-dimensional microstructure was characterized by scanning electron microscopy (SEM, FEI Scios) and electron back scattering diffraction (EBSD, Oxford Nordlys Max 3). Energy dispersive spectroscopy mapping (EDS, Oxford X-Max80T) was employed in the scanning transmission electron microscopy (STEM, JEOL 2100 F) mode for composition estimate. The phase constitute of the bulk composite was probed by X-ray diffractometer (Bruker D8 Advance, Da Vinci, Cu Kα). The phase constitution was analyzed in Jade (version 6.5) via Rietveld refinement. In the X-ray diffraction (XRD) patterns, the tetragonal (t) phase was inspected via (101) peak at -30° and the monoclinic (m) phase via (111) peak at -28° and (11$\bar{1}$) peak at -31°. Specific peak location is slightly affected by internal and external stresses. The weight fraction of monoclinic phase $X_m$, tetragonal/cubic phase $X_{t/c}$ could be expressed by the following equations[46],

$$X_m = \frac{I(11\bar{1})_m + I(111)_m}{I(11\bar{1})_m + I(111)_m + I(101)_{t/c}} \qquad (2)$$

$$X_{t/c} = 1 - X_m \qquad (3)$$

where $I(111)_m$, $I(11\bar{1})_m$, $I(101)_{t/c}$ are the integrated intensity from the monoclinic (111), (11$\bar{1}$) peaks, the tetragonal/cubic (101) peak, respectively. XRD conducted at room temperature had a 2θ range of 20–90° with a scan speed of 5°/min. The in-situ high-temperature XRD is equipped with accessories HTK 2000N. The temperature step size was set to be 50 °C with a ramping rate of 50 °C/min. A holding segment of 5 min was carried out at each temperature to achieve thermal equilibrium, followed by XRD scanning in the 2θ range of 27–32° at a scan speed of 2°/min. Uniaxial compression tests on the bulk composite and the pure Al (2 mm-diameter and 4 mm-height cylindrical-shaped) were conducted by an INSTRON 3340 tester at a constant strain rate of $5 \times 10^{-4}$ s$^{-1}$. True stress-strain curves were refined by subtracting the substrate effect to reduce the deformation effect of the compression plate. Uniaxial compression tests on the micro-pillars were conducted at a strain rate of $5 \times 10^{-4}$ s$^{-1}$ on the KLA G200 nanoindenter equipped with a 10 μm-diameter flat punch diamond tip with continuous stiffness measurement. True stress-strain curves were employed to characterize the deformation behavior following the methodology raised by Greer et al.[24]. Five samples were used for each bulk and pillar compression test. Post-compression micro-pillars were milled into TEM cross-section lamellas by FIB, followed by successive annealing at 200 °C, 400 °C, and 600 °C in argon atmosphere, with a ramping rate of 5 °C/min and holding time of 5 min. Phase distribution in the cross-section was characterized via precession electron diffraction (PED, NanoMEGAS) that is attached to TEM, operated at

200 keV, with a spot size of ~1 nm and a step size of 10 nm. The collection and analysis of diffraction patterns with a precession angle of 0.6° were conducted using the ASTAR™ automated crystal mapping system.

## Data availability

The authors declare that the data supporting the findings of this study are available within the paper and its supplementary information files. The source data underlying Fig. 3a, b are provided as a Source data file. Source data are provided with this paper.

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

## Acknowledgements

The authors would like to thank Dr. Zehui Du from Nanyang Technological University (NTU) and Prof. Xuejun Jin from Shanghai Jiao Tong University (SJTU) for their valuable discussions. Dr. Yu Liu, Dr. Yuyang Liu, and Dr. Yifei Peng from (SJTU) are also acknowledged for their help with the TEM specimen preparation and PED operation. This work is financially supported from the National Key R&D program of China (2022YFB3705704), and the National Natural Science Foundation of China (Nos. 52192595, D.Z. and 52001204, L.Z.).

## Author contributions

C.G., D.Z., and Q.G. were the lead scientists of the study and proposed the core concept; Y.S. and W.Z. designed and fabricated the sample; S.J. supported in pillar fabrication and mechanical experiment test; L.L. carried out the XRT and XRD experiments and analysis; W.Z. performed the SEM, TEM investigation (TEM/STEM/EDS/PED) and wrote the paper. L.Z. and Q.G. supervised the research. All authors contributed to the discussion.

## Competing interests

The authors declare no competing interests.
