## [Peer Review File · Nature Communications]

Realizing reversible phase transformation of shape memory ceramics constrained in aluminumReviewers' comments:

Reviewer #1 (Remarks to the Author):

Review of the manuscript Realizing reversible phase transformation of shape memory ceramics constrained in aluminum by Zheng, W. et al.

The authors present interesting results of a single micropillar experiment and draw far-reaching conclusions. Moreover, the geometric distribution of ceramic particles in that single micropillar is a peculiar one: an arch of ceramic particles from upper left, through the center and back to lower left. In principle, all the authors' conclusions may be direct consequence of this particular realization of composite geometry. Other geometries may produce cracks, dislocations may not anneal,...

Therefore, current results are not fit for publication. Further experimentation is needed:

(1) As the very minimum, establish repeatability of results (mechanical load, reversibility upon annealing and absence of cracks). Minimum three realizations (micropillars) if they the results turn out to be consistent, more if they are not!

(2) Document variability (or constancy) in microstructure (geometry of ceramic particles in the Al matrix). If the results are consistent for variable microstructures, only then one may theorize about mechanisms (dislocation annealing, strong interface,...). If the microstructures are similar to the arch described above, then the shape of the reinforcement must feature in the explanation.

(3) Finally, the argument about reversibility of dislocation motion begs the question: What if you reload the same pillar? Will it also anneal without cracking?

Reviewer #2 (Remarks to the Author):

"Realizing reversible phase transformation of shape memory ceramics constrained in aluminum". Detailed comments on the manuscript can be found below.

The paper is certainly of interest and deserves publication although some problem areas in the text first need attention.

The authors appear to have come close to their goal of integrating shape memory ceramics (SMCs) into a matrix material and realizing their reversible tetragonal-monoclinic phase transformations without destructive impact. However, if they really want to advertise their research as a breakthrough, they should do some more experiments, in particular on the effects of multiple thermal cycles on the material as well as the effect of multiple mechanical load cycles (fatigue).

Detailed comments:

Page 3 line 36: "a reversible deformation strains" should read : a reversible deformation strain

Page 4 line 53: "the martensitic transformation behavior of the constrained shape memory ceramics subjected to thermal and stress stimuli were examined". Yes but these are not sufficient to confirm the usefulness of the Al-CZ material.

Page 5 line 75: " all the monoclinic phases in the CZ/Al composite were found to transform to the tetragonal phase with 100% relative density" should read "all the monoclinic particles in the CZ/Al composite, with 100% relative density, were found to transform to the tetragonal phase."

Page 7 line 96: "(Fig. 1b and Fig. S1)." Reference to Fig S1 is not appropriate here since Fig S1 is solely concerned with the CZ particles not the composite.

Page 8, figures 2d and 2e: It is clear from 2d that the CZ particles are not single crystal but agglomerates. Yet also elsewhere in the manuscript they are referred to as CZ crystals. In the Method section (page 20 line 312) it is claimed that they were synthesized in single crystal form and. that may be so, but it is clear that agglomeration occurred during processing of the composite.

Page 8 line 111 one mentions "band contrast". I suppose Kikuchi bands are referred to. Perhaps one should write "crystal orientation contrast" On the same line: what does IPZ mean?

Page 9 line 127: "Such a proposal is corroborated by the absence of diffraction peak intensities of the tetragonal phase in the vicinity of $\sim 500^{\circ}\text{C}$ (marked as "gap" in Fig. 3b), a clear evidence of the long-range disordering of the Al matrix due to the softening at a temperature near the melting point, which may cause noises that covered/ blurred the diffraction peaks of the CZ particles.ref23". This paragraph is pure nonsense. 500°C is still 160°C below the melting point of Al. As the melting point is approached some peak broadening of aluminum may occur but there should still be clear peaks present even within a few degrees of the melting point. What does long range disordering mean? Crystallographic order is lost and the aluminum melts far below its melting point? What does softening mean here in the context of diffraction of X-rays? Ref 23 is not helpful.

I think the " gap" in Fig 3b is probably related to an instrumental problem such as a diffractometer temporarily going out of alignment.

Page 11 line 145:" Strikingly, the tetragonal phase ... showed a slight reduction to 94 wt.% in the annealed composite (Fig. 3c)." When considering this result, one wonders how the composite material would behave when multiple thermal cycles are applied as would be the case if a true shape memory effect is to be exploited.

Page 11 line 151:" Compressive response of the bulk composites" A compressive load will, at least during initial loading, further support the Al matrix constraint to prevent the phase transformation to monoclinic in the CZ particles. Why was tensile loading of the bulk composite not considered to test the stability of the tetragonal phase and the effectiveness of the aluminum matrix constraint?

Page 14 line 210:" from $\sim 38.7\%$ in the deformed state to $\sim 48.7\%$ in the 200°C -annealed state". Is the last figure in these results significant?

Page 15 Fig. 5: One may question the relevance of the results shown in this figure. Annealing was carried out on a very thin foil where the constraint of the aluminum matrix is only present in two dimensions, at best. How relevant are these observations then for the actual composite material?

Page 19 line 308:" These findings suggest the possibility of realizing multiple cycles of shape memory effect or superelasticity in the CZ/Al composites for future structural and functional applications."

Experiments on multiple thermal cycles and multiple mechanical cycles preferably in tension or at least

in bending mode are indeed missing in this paper in order to claim a true breakthrough.

Page 20 line 319: "powders ("as-milled CZ/Al powders") were put into a die". One of the most interesting results of the work is the interface between the CZ particles and the aluminum matrix which was found to be a thin amorphous layer of Al_2O_3 . The processing conditions may play an important role and all details could be important. I suppose the die material is steel considering the very high compressive stresses used for densification? Were there any coatings or foils used to prevent sticking of the composite to the die material?

Fig. S1 (b) In this important graph the horizontal temperature scale is not clear. Where is the vertical line corresponding to 500C. For the determination of the A_s A_f and M_s temperature presumably tangent constructions were used?

Fig. S1 (c) also here a temperature scale would be useful

Reviewer #1: The authors present interesting results of a single micropillar experiment and draw far-reaching conclusions. Moreover, the geometric distribution of ceramic particles in that single micropillar is a peculiar one: an arch of ceramic particles from upper left, through the center and back to lower left. In principle, all the authors' conclusions may be direct consequence of this particular realization of composite geometry. Other geometries may produce cracks, dislocations may not anneal.... Therefore, current results are not fit for publication. Further experimentation is needed.

Response: We thank the reviewer very much for his/her insightful comments and questions over our study. In fact, we have conducted multiple tests on several sets of samples and only the representative results are shown in the as-reviewed manuscript. Following the reviewer's comments and suggestions, we have done more systematic experimentation to show the reproducibility and reliability of the data to fully address the questions raised. Please refer to our detailed responses as follows.

1. As the very minimum, establish repeatability of results (mechanical load, reversibility upon annealing, and absence of cracks). Minimum three realizations (micropillars) if they the results turn out to be consistent, more if they are not!

Response: We thank the reviewer very much for this suggestion. We have done the following additional experiments to verify the repeatability and results.

1) First, we conducted a high-resolution X-ray computed tomography microscopy (XRM) analysis to obtain an overview on the spatial distribution of cerium-doped zirconia (CZ) particles in the aluminum (Al) matrix (**Fig. R1**). As shown, a generally inter-penetrated distribution of the two constituent materials (Al and CZ) in the whole analyzed area is observed. The magnified area with

highlighted Al grains (red) surrounded by CZ particles (blue) further illustrates the interpenetrating nature and spacial network structure of the CZ particles in the Al matrix (**Fig. R1b**, note: the CZ particles in front of the Al grains were purposely removed to ensure a clearer view).

Fig. R1 a High-resolution X-ray computed tomography microscopy (XRM) volume renderings of CZ/Al bulk composite, with CZ particles (blue) and Al matrix (grey). **b** The representative magnified area with highlighted aluminum (Al) grains (red) surrounded by CZ particles (blue). The CZ particles in front of the Al grains were purposely removed to ensure a clearer view.

2) Thanks for reminding us about the consistency of our data from a statistical point of view. In addition to the micro-pillar (denoted as S1 here) already presented in the initial manuscript, two more micro-pillars (S2, S3) fabricated from different locations of the composite are added in **Fig. 4c** of the revised manuscript. The effective CZ particle contents in pillars S1-S3 were approximately 30 vol.%, 67 vol.%, and 70 vol.%, respectively, estimated by the area fractions of CZ and Al in the TEM cross-section. The spatial distribution of CZ particles in pillar S1 was in an arc-like arrangement, as suggested by the reviewer, while in pillars S2 and S3 the particles spread over most of the cross-section areas. Although the particle assembly patterns were different in the 3 pillars, most of the CZ particles were interconnected with each other, forming clusters embedded in the Al matrix. This ensured effective load transfer from the Al matrix to different particles and as a result, the mechanical response and plastic flow behavior of all the 3 pillars were found to be similar, despite the variation in their compressive strengths owing to the difference in the CZ particle content/distribution in each pillar (**Fig. R2a**). The pillars S1-S3 were thus used to test the repeatability of the results from mechanical loading and for reversibility tests upon annealing.

The pre- and post-compression morphologies of the pillars S1-S3 are shown in **Fig. R2b-d**, where the pillars S2, S3 after compression experienced severe plastic deformation on their lateral surface (evidenced by the bulges due to local extrusion of the Al matrix), which are analogous to the deformation behavior of micro-pillar S1. **These results thus prove the repeatability of the mechanical behavior of the pillars under compressive loadings.**

Fig. R2 *a* True stress-strain curves of CZ/Al micro-pillars S1, S2, S3. *b-d* SEM image of pre-compression (left) and post-compression (right) CZ/Al micro-pillars *b* S1, *c* S2, and *d* S3. *e-g* TEM cross-section of *e* S1, *f* S2, *g* S3. *h-j* Precession electron diffraction (PED) bright field (BF) images, and the phase mappings on the TEM cross-section lamella of micro-pillars: *h* S1, *i* S2, and *j* S3 and post-compression state (left) and the 400°C annealed state after compression (right).

Stress-induced martensitic transformation was further confirmed at different stress levels within micro-pillars S2, and S3 and the results are shown in **Fig. S12** in the revised manuscript and reproduced in **Fig. R2h-j** above. The CZ particles in all three pillars have been shown to experience martensitic transformation under compression. The highest stress-induced monoclinic phase content of ~68% was observed in the post-compression pillar S3 according to the precession electron diffraction (PED) analysis, which exhibited the highest compressive strength (5% compressive flow stress ~545 MPa) among all three pillars. After 400°C annealing, more than half of the stress-induced monoclinic phase

content (50%, and 79%, respectively) in pillars S2 and S3 were converted back to the tetragonal phase, comparable to that of the S1 pillar (~68.9%, as shown in our initial manuscript). **Therefore, although the local CZ content (and subsequently the micro-pillar strength) may vary, as long as an interconnected, or “force-chain” configuration¹ of the particles is formed across the entire pillar gauge length for effective load transfer, a substantial fraction of CZ particles would be able to transform to the monoclinic phase upon loading and such phase transformation is reversible at the subsequent 400°C annealing.** This phenomenon has been observed in the three pillars, S1-S3, and therefore, is repeatable and representative to the phase transformation behavior of the CZ particles under stressing or annealing. More details on the requirements for successful phase transformation are provided in our response to Reviewer #1, Comment #2 in the following.

In the revised manuscript, to address the repeatability of our results, we have added the results of XRM, and the data for micro-pillars S2-S3 in **Fig. 1**, and **Fig. 4**, respectively. We also provided the PED data for micro-pillars S2 and S3 in the revised supplementary materials. It is worth mentioning that the new data further verify the reversible martensitic transformation in the CZ particles, and therefore the relevant discussions/conclusions made in the manuscript remain valid.

2. Document variability (or constancy) in microstructure (geometry of ceramic particles in the Al matrix). If the results are consistent for variable microstructures, only then one may theorize about mechanisms (dislocation annealing, strong interface, ...). If the microstructures are similar to the arch described above, then the shape of the reinforcement must feature in the explanation.

Response: We thank the reviewer very much for his/her reminder and we fully agree that the

microstructures have a strong influence on the martensitic transformational behavior. Studying the samples with different microstructures is necessary for us to theorize the controlling factors responsible for the phase transformation behavior in the CZ particles in the CZ/Al composite pillars.

For this purpose, we have selected pillars S1-S3 as representative samples to study. As mentioned above, the CZ particle contents in pillars S1-S3 were approximately 30 vol.%, 67 vol.%, and 70 vol.%, respectively. The spatial distribution of CZ particles in the pillar S1 was in arc-like arrangement, as suggested by the reviewer, while in pillar S2 and S3 the particles were interconnected with each other (the pattern in S3 resembled a “double-arc” feature, as shown in **Fig. R2g** and **Fig. R2j**). Although pillars S1-S3 were different in the CZ particle content and assembly patterns, they generally had the particles forming a network/force-chain configuration which allowed for effective load transfer between the matrix and the particles, and thus up to 68% particles experienced martensitic transformation under compression.

To further confirm the critical factor for stress-induced martensitic transformation, we also intentionally fabricated a pillar S4 with discretely distributed particle clusters *without* a force-chain configuration. In particular, in pillar S4, three *unconnected* CZ particle clusters were laminated with Al grains across the pillar gauge length (**Fig. R3**), in stark contrast from the geometries of pillar S1-S3 (S1: arc as the reviewer pointed out, S2: interconnected clustering, S3: double arc). Herein, a prominent shear occurred in the post-compression pillar S4 (**Fig. R3a, b**), leading to a totally different deformation mode and significantly lower compression strength (~180 MPa), even at an apparently high CZ volume ratio of ~55 vol.%. Furthermore, we also observed a much smaller fraction of stress-induced martensite phase formed in the pillar after compression (~13%, **Fig. R3b**), indicating a strong correlation between the spatial distribution of CZ particles in the pillar and the extent of phase

transformation upon mechanical loading. After annealing at 400°C, 62% of the transformed particles in pillar S4 were converted back to the tetragonal phase, featuring a similar thermal response as pillars S1-S3 (**Fig. R3c**).

Through the studies on S1-S4 pillars with varying microstructures, we can then conclude that the spatial distribution of CZ particles in a force-chain configuration is critical to obtain high conversion rate of stress-induced martensite phase in the CZ/Al composite pillars. We have added the new data of S4 pillar in the revised supplementary materials (**Fig. S8, 12, 14**), and a thorough discussion on the role of microstructure on the stress-induced phase transformation is given in the revised manuscript (Page 17-18).

Fig. R3 a True stress-strain curves of CZ/Al micro-pillar S4. **b-c** PED bright field (BF) image and the phase mapping on the TEM cross-section lamella of micro-pillar S4: **b** the post-compression state and **c** the 400°C annealed state after compression.

From the view of the existing literature raised by Yu et al.^{1, 2, 3}, the martensitic transformation was prominently induced in granular shape memory ceramics, with force network at the macroscale and stress concentration at the microscale (**Fig. R4a, b**). This involves a highly heterogeneous

distribution of the driving force and very low mechanical constraint for martensite nucleation. The observation in this work has further confirmed and extended the mechanism of martensitic transformation from granular shape memory ceramics to bulk samples:

- i. **3-Dimensional network of particles.** The CZ/Al composites possessed a three-dimensional network structure of the particles (**Fig. R4d**), with shape memory ceramic submicron particles ($\sim 0.5 \mu\text{m}$) preferably located at the Al grain boundaries.
- ii. **Force-chain configuration.** Our study provided a broad choice for designing multiple geometries within micro-pillars to construct the force chains and subsequently realize the reversible martensitic transformation (**Fig. R4e, f**).

Fig. R4 *a* Illustration of the co-evolution of packing microstructure, lattice strain, and phase constitution during compression. *b* Diagram of martensite plate formation in granular packings during

deformation.³ *c* SEM overview of a CZ/Al composite cross-section, showing the typical network structure in 2D projection. *d* XRM volume renderings of CZ/Al composites, reproduced from **Fig. R1**. *e* Overview of as-milled micro-pillars located in regions populated with CZ particles. *f* The representative micro-pillars B1-B4 with different CZ contents and connections, where the microstructure of pillar S4 resembles B2, and pillars S1-S3 resembles B4.

In summary, **the particle network/force-chain configuration was found to be a must for effective load transfer and the ensuing phase transformation in the composites.** The CZ particle content and distribution in the pillar, and the mechanical and transformational behaviors of micro-pillars S1 to S4 are summarized in **Table S3** in the revised supplementary materials, which are reproduced here as **Table R1**. We also added one figure to theorize about mechanisms in the revised supplementary materials (**Fig. S15**, reproduced here as **Fig. R5**). The mechanisms of reversible martensitic transformation in CZ/Al without causing fracture could be attributed to the strong geometric confinement offered by the Al matrix, the robust CZ/Al interface, the localized three-dimensional particle network/force-chain configuration that effectively transferred mechanical loads, and the decent flowability of the matrix that accommodated the volume change during phase transformation.

Fig. R5 Schematic mechanism of full austenization and reversible martensitic transformation in shape memory ceramics constraint in Al. **a** Monoclinic zirconia and Al. **b** Tetragonal zirconia stabilized by the matrix constraint. **c** 3D architecture of as-fabricated composites. **d** A representative local region with densely populated zirconia particles. **e** Stress-induced martensitic transformation and **f** thermally induced reverse martensitic transformation. The interfacial structure corresponding to **d-f** are magnified in **g-i**, where the white arrows correspond to the direction of volume expansion/contraction. The textboxes suggest the structure and property evolution during the thermo-mechanical treatments, i.e., the reversible phase transformations.

Table R1 Summary of CZ particle content and distribution in the pillar, and the mechanical and transformational behaviors of micro-pillars S1 to S4.

	Micro-pillar	S1	S2	S3	S4
Nature of CZ particles	CZ Distribution	Arc	interconnected clustering	Double arc	Laminated
	Force-chain?	Yes	Yes	Yes	No
	CZ Content/vol. %	30	67	70	55
Mechanical Behavior	Compressive Strength/MPa (5% flow stress)	391	282	545	180
	Energy absorption/ MJ·m ⁻³	54.0	37.5	74.8	24.8
	Deformation mode	Load transfer through CZ chains local extrusion of Al			Al shear
Transformational Behavior	Mechanically transformed fraction (Mono. ratio)	61%	30%	68%	13%
	Mono. ratio after 400°C	31%	15%	14%	5%
	Thermally transformed fraction (Tetra. ratio)	30%	15%	54%	8%
	Reversible transformation?	Yes	Yes	Yes	Yes

3. Finally, the argument about the reversibility of dislocation motion begs the question: What if you reload the same pillar? Will it also anneal without cracking?

Response: We thank the reviewer very much for his/her constructive questions. As the reviewer correctly pointed out, it would be interesting to reload the same pillar and examine the corresponding microstructural evolution. However, in order to verify the stress-induced martensitic transformation, all the post-compression pillars were milled into TEM lamellas. Therefore, after TEM observation, we were not able to reload and anneal the same pillar under exactly the same loading and thermal treatment parameters.

In order to address the reviewer's question, a new set of three pillars (C1-C3) have been milled and went through different loading-annealing-reloading cycles (**Fig. R6a**), with the purpose of better comparing and revealing the evolution of dislocations. Specifically, C1 was compressed only once without post-compression annealing. C2 was compressed and then annealed at 400°C. C3 went through a compression-400°C annealing-compression cycle. The representative dislocation distributions are shown in **Fig. R6b**. As manifested, after the first and second loadings, dislocations in the Al matrix were gathered (also indicated by geometric phase analysis in **Fig. R6c**), which almost disappeared after 400°C annealing. These observations undoubtedly confirm the reversibility of the dislocation motion.

From low-magnification bright field images taken at TEM cross-sections (**Fig. R6b**), there was no obvious cracks in all the micro-pillars C1 to C3, indicating that decent flowability of the Al matrix may accommodate the volume change during martensitic transformation of CZ particles, as we summarized in the response to Reviewer #1, Comment #2. In other words, the pillars generally maintained their structural integrity under cyclic stimuli (reloading the same pillar and then annealing),

which is consistent with the conclusions made in the manuscript.

Fig. R6 *a* Representative SEM image of micro-pillars C1-C3 under stress-thermal cycles. *b* TEM analysis on the low-magnification morphology, dislocations of the post-mortem micro-pillars in *a*. *c* Geometric phase analysis (GPA) analysis on micro-pillar C3, revealing the distortion of the lattice.

Reviewer #2: The paper is certainly of interest and deserves publication although some problem areas in the text first need attention. The authors appear to have come close to their goal of integrating shape memory ceramics (SMCs) into a matrix material and realizing their reversible tetragonal-monoclinic phase transformations without destructive impact. However, if they really want to advertise their research as a breakthrough, they should do some more experiments, in particular on the effects of multiple thermal cycles on the material as well as the effect of multiple mechanical load cycles (fatigue):

Response: We thank the reviewer very much for his/her positive evaluation on our manuscript, and have done our best to address the reviewer's questions/comments.

1. Page 3 line 36: "a reversible deformation strains" should read: a reversible deformation strain.

Response: We greatly appreciate the reviewer's careful examination. Following the reviewer's comment, we have corrected this sentence and also carefully checked the grammar throughout the revised manuscript.

2. Page 4 line 53: "the martensitic transformation behavior of the constrained shape memory ceramics subjected to thermal and stress stimuli were examined". Yes, but these are not sufficient to confirm the usefulness of the Al-CZ material.

Response: This is a very good point and we thank the reviewer very much. It must be recognized that there is still a distance from research stage to application of the CZ-Al composites. In this work, the CZ-Al composites are served as model material system, with the purpose of demonstrating the mechanically or thermally induced martensitic transformation of the CZ particles under a constraint

from a soft metal (Al) matrix. Based on the underlying mechanisms and associated criteria for reversible phase transformations in the composite, one could further explore the application of transformation-induced plasticity (TRIP)⁴ and elastocaloric effect⁵ in these composite with controllable thermal and stress stimuli. Moreover, the potential technical significance of this study may be appreciated from the following aspects:

- 1) From the perspective of the metal matrix, the feasibility of martensitic transformation under the geometrical constraint of Al inspires further research to integrate shape memory ceramics into other matrices (i.e., high strength steels, engineering Al alloys, and etc.), to form composites for structural and intelligent applications.
- 2) From the perspective of shape memory ceramics, as discussed in the *Introduction* of the manuscript, the biggest technical hurdle for their application is to scale up their dimension without causing catastrophic failure during mechanical loading⁶. The incorporation of a soft matrix (Al in our study), as the medium to transfer load and accommodate the volume change upon phase transformation of the ceramics, renders a feasible route to obtain bulk composites retaining the features of shape memory ceramics.

3. Page 5 line 75: “all the monoclinic phases in the CZ/Al composite were found to transform to the tetragonal phase with 100% relative density” should read “all the monoclinic particles in the CZ/Al composite, with 100% relative density, were found to transform to the tetragonal phase.”

Response: We greatly appreciate the reviewer’s careful examination on the language and readability of our manuscript. We have made the correction following the reviewer’s suggestion, and we have also done a thorough inspection on the language of the manuscript to improve its quality.

4. Page 7 line 96: “(Fig. 1b and Fig. S1).” Reference to Fig S1 is not appropriate here since Fig S1 is solely concerned with the CZ particles, not the composite.

Response: We thank the reviewer very much for his/her reminder, and we greatly appreciate his/her careful examination on the figures. In the revised supplementary materials, we have corrected it to be **Fig. S5**.

5. Page 8, figures 2d and 2e: It is clear from 2d that the CZ particles are not single crystal but agglomerates. Yet also elsewhere in the manuscript they are referred to as CZ crystals. In the Method section (page 20 line312) it is claimed that they were synthesized in single crystal form and. that may be so, but it is clear that agglomeration occurred during the processing of the composite.

Response: We thank the reviewer very much for pointing this out. The reviewer correctly points out that the CZ particles existed in the composite as agglomerations of single crystals. In particular, the CZ particles were synthesized via a gel-casting method⁷, during which the ball milling (200 rpm for 8 h in ethanol) was conducted to completely separate the CZ grains into single crystals with a size of ~0.5 μm . The small size and high surface energy of the particles make them easily agglomerate, so that the as-fabricated CZ particles were essentially clusters of single crystals. Subsequent co-milling with the Al powders and the following densification processing steps rendered a distribution of particles that preferably located at the Al grain boundaries, forming a network structure (further confirm in 3D architecture by XRM in **Fig. R1** and **Fig. 1d** of the revised manuscript). In fact, the agglomeration (or clustering) of the CZ single crystals in the composite allowed effective load-transfer via the Hertzian contact and was proved to be crucial for the achievement of stress-induced martensitic transformation

(please refer to our responses to Reviewer #1, Comments #1, and #2 for details).

Following the reviewer's comment, to make the description more precise, we have replaced "CZ single crystals" to "CZ single-crystal clusters" when describing the particles in the composite in the revised manuscript.

6. Page 8 line 111 one mentions "band contrast". I suppose Kikuchi bands are referred to. Perhaps one should write "crystal orientation contrast" On the same line: what does IPZ mean?

Response: We thank the reviewer very much for his/her reminder and advice. We are sorry for the misspelling, which should actually be "IPF Z" (inverse pole figure at Z axis). In the revised manuscript, the sentence has been revised to be "Electron back scattering diffraction (EBSD) of CZ/Al and pure Al, mapped with Kikuchi band contrast and crystal orientation contrast (Z axis)".

7. Page 9 line 127: "Such a proposal is corroborated by the absence of diffraction peak intensities of the tetragonal phase in the vicinity of ~500°C (marked as "gap" in Fig. 3b), a clear evidence of the long-range disordering of the Al matrix due to the softening at a temperature near the melting point, which may cause noises that covered/ blurred the diffraction peaks of the CZ particles. Ref 23". This paragraph is pure nonsense. 500C is still 160C below the melting point of Al. As the melting point is approached some peak broadening of aluminum may occur but there should still be clear peaks present even within a few degrees of the melting point. What does long range disordering mean? Crystallographic order is lost and the aluminum melts far below its melting point? What does softening mean here in the context of diffraction of X-rays? Ref 23 is not helpful. I think the " gap" in Fig 3b is probably related to an instrumental problem such as a diffractometer temporarily going out of

alignment.

Response: We thank the reviewer very much for his/her comment and advice. To validate our results, we repeated the high-temperature XRD test (**Fig. R7a**). Indeed, as shown in **Fig. R7b**, the “gap” was no longer there. We also checked the Al (111) peak and found that the Al still maintained a clear peak at 500°C with peak shift (caused by thermal expansion) in **Fig. R7c**. Therefore, the previous “gap” in the XRD spectrum was likely to be a fluke and, as the reviewer suggests, may be the result of an instrumental problem. Following the reviewer’s comment, in the revised manuscript, we have deleted the related sentence and revised the figure (**Fig. 3b**). It is worth mentioning that the new batch of data doesn’t change the thermal response of the composite, nor affect the subsequent discussions/conclusions made in the manuscript.

Fig. R7 a X-ray diffraction spectrums obtained from in-situ high-temperature XRD measurement. **b** 2θ-temperature-intensity contour map of tetragonal zirconia phase obtained from **a**. The area representing Al (111) peak was present in **c**.

8. Page 11 line 145:” Strikingly, the tetragonal phase ... showed a slight reduction to 94 wt.% in the annealed composite (Fig. 3c).” When considering this result, one wonders how the composite material would behave when multiple thermal cycles are applied as would be the case if a true shape memory effect is to be exploited.

Response: We thank the reviewer for the constructive suggestions. Following the reviewer’s advice, we have done multiple (15) annealing (500°C)-cryogenic (-196°C, liquid nitrogen) test cycles to further reveal the role of Al matrix constraint, and the results have been added in **Fig. 3e** of the revised manuscript. Contrary to expectations, the tetragonal phase ratio basically kept unchanged (around 94 wt.%) during the following 15 thermal-cryo cycles (reproduced here as **Fig. R8a, b**). In addition, all of the characteristic peaks had a more broadened full-width at half-maximum (FWHM, from 0.11° to 0.26°) and were slightly shifted from 38.42° to 38.82° with increasing number of cycles (**Fig. S7b** and **Table S2** in the revised supplementary materials, reproduced here as **Fig. R8c, d**). According to the Williamson-Hall method⁸, such a trend was likely to originate from the increment of dislocation density during the thermal-cryo cycles (from $3.82 \times 10^{12} \text{ m}^{-2}$ to $8.63 \times 10^{12} \text{ m}^{-2}$), which further inhibited the martensitic transformation and stabilized the tetragonal phase.

Finally, we would like to mention that our ongoing work includes the detailed study of the thermodynamic and kinetic parameters for the stabilization of the tetragonal phase and the reversible phase transformation of CZ particles in a geometrically constrained state. These results are out of the scope of this study and will be published in another paper in the future.

Fig. R8 *a* XRD spectrums of the CZ/Al composite under multiple annealing - cryogenic treatments. Normalized Al (111) peaks are magnified in *b*. The corresponding tetragonal phase ratio, full-width at half-maximum (FWHM), and the dislocation density are shown in *c*, *d*. (Notes: C0 - as-fabricated CZ/Al; C1, C5, C10, C15 - CZ/Al composite after 1,5,10,15 cycles of thermal treatment.)

9. Page 11 line 151: "Compressive response of the bulk composites" A compressive load will, at least during initial loading, further support the Al matrix constraint to prevent the phase transformation to monoclinic in the CZ particles. Why was tensile loading of the bulk composite not considered to test the stability of the tetragonal phase and the effectiveness of the aluminum matrix constraint?

Response: We thank the reviewer for this important question, and fully agree that, generally, a compressive load will, at least during initial loading, further support the Al matrix constraint to prevent

the phase transformation to monoclinic in the CZ particles. As a matter of fact, the selection of the compression loading mode was based on the following two considerations:

- 1) **Ease of phase transformation under compression.** Firstly, the martensitic transformation was triggered by a stress level equal to or higher than the critical resolved shear stress^{2, 6, 9, 10, 11, 12}. As we stated in our previous responses to Reviewer 1, Comments #1, and #2, the particle network/force-chain configuration was found to be a must for effective load transfer and the ensuing phase transformation in the composites. Along with this line of reasoning, effective load transfer between CZ particles and CZ particle/Al matrix can be more easily realized in the compression mode, than in the tensile or bending mode, both of which are sensitive to the micro-cracks, and may thus make the composite subject to early failure. Furthermore, the achievement of the critical resolved shear stress for phase transformation on the CZ particles was also confirmed by the rule-of-mixture and the Hertzian contact model, as discussed in the *Discussion* part of the revised manuscript.
- 2) **Limited tensile strain on the composites.** Following the reviewer's comment, we have done micro-tensile test as well. The test results of three specimens (T1, T2, and T3) are shown in **Fig. R9**. Despite high tensile strength (~300 MPa) for this set of samples, unfortunately, the tensile ductility was fairly low (uniform elongation 2-4%, total elongation 3-5%). This may be the result of the inhomogeneous distribution of CZ particles in the matrix. Therefore, it was difficult to use tensile specimens to carefully study the flow behavior of the matrix, where the lack of extended plastic deformation of the matrix may prevent appreciable phase transformation.

Fig. R9 Micro-tensile test of as-fabricated CZ/Al composites. **a** Pre- and post-tension SEM images of T1, T2, and T3 samples. **b** True stress-true strain curves of the three samples in **a**.

We would like to mention that, in our ongoing study, we have adjusted the fabrication parameters and obtained the CZ/Al composites with uniformly dispersed CZ particles in the Al matrix. The tensile stress-strain curves in **Fig. R10a** show that the composites had both enhanced tensile strength (+67.1%, from 151.8 ± 3.3 MPa to 253.6 ± 5.1 MPa) and ductility (+28.2%, from 7.1 ± 0.1 % to 9.1 ± 0.3 %) over those of the unreinforced matrix, achieving an excellent strength-ductility synergy. In addition, XRD on the necking region indicates that the CZ particles were partially transformed into a monoclinic phase under tensile stress loading (**Fig. R10b**), which was also confirmed by the SAED pattern (**Fig. R10c, d**). These results are out of the scope of the present study and will be reported in a paper in future.

Fig. R10 a Engineering tensile stress-strain curve of CZ/Al composites and pure Al by a new set of fabrication parameters, with the corresponding SEM images taken at the fracture surfaces shown as the insets. **b** XRD spectrum of the necking region in the post-tension CZ/Al composites, showing distinct monoclinic peaks. **c** TEM image of post-tension CZ/Al composites, and the SAED taken on a representative CZ particle is present in **d**.

10. Page 14 line 210: "from ~38.7% in the deformed state to ~48.7% in the 200°C-annealed state". Is the last figure in these results significant?

Response: We fully agree with the reviewer that, the last figure of the phase ratio is insignificant. They have been deleted from the PED mapping in the revised manuscript (so these numbers become 39% and 49%, respectively).

11. Page 15 Fig. 5: One may question the relevance of the results shown in this figure. Annealing was carried out on a very thin foil where the constraint of the aluminum matrix is only present in two dimensions, at best. How relevant are these observations then for the actual composite material?

Response: We thank the reviewer for raising this up. This was indeed a point we carefully examined in the experiments. Our results and associated discussions are formulated as follows:

- 1) As shown in **Fig. 2** in the revised manuscript (reproduced here as **Fig. R11a**), the tetragonal phase was retained in the TEM lamella of the as-fabricated CZ/Al composite. It is found that the relaxation of the geometrical constraint in one dimension (i.e., the thickness direction) has a negligible impact on the stability of the tetragonal phase in the as-fabricated composite, as the phase constitution observed from TEM foils was consistent with that obtained from the bulk samples. Therefore, the TEM results can be used to interpret the evolution of constituent phases during subsequent thermo-mechanical treatments.
- 2) In this study, reversible martensitic phase transformation has been observed in the micro-pillar and TEM lamella thereof. On the other hand, we only observed a very small monoclinic peak intensity in the compressed bulk sample by XRD (**Fig. S10** in the revised supplementary materials, reproduced here as **Fig. R11b**). Such a discrepancy between the two length scales may be rationalized by the following two aspects: i) XRD on the bulk sample can only detect the surface layer, so whether martensitic transformation happened or not in the sample interior is not detectable by XRD. ii) the global CZ content (~16 vol.%) in bulk composite may not be high enough for sufficient load transfer required for transformation, as compared to the localized micro-pillars with high CZ content (>30 vol.%, as we depicted in **Fig. R2**).
- 3) Following these observations, in our ongoing study, XRD verified phase transformation in the

necked region of post-deformation bulk tensile samples (please refer to our previous responses to Reviewer 2, Questions #9 and **Fig. R10b**, reproduced here in **Fig. R11c**). After heat treatment on the post-tension sample, the tetragonal phase increased from 71 wt.% to 85 wt.%. In other words, reversible phase transformation was achieved in bulk composite samples, which is consistent with the observation in TEM lamella.

Fig. R11 a High-resolution TEM analysis of the interface between CZ and Al with fast Fourier transformation analysis (FFT) and IFFT. XRD spectrums of **b** bulk CZ/Al after uniaxial compression, **c** XRD spectrums of CZ/Al composites after tension and annealing by a new set of fabrication parameters.

12. Page 19 line 308:” These findings suggest the possibility of realizing multiple cycles of shape memory effect or superelasticity in the CZ/Al composites for future structural and functional applications.” Experiments on multiple thermal cycles and multiple mechanical cycles preferably in

tension or at least in bending mode are indeed missing in this paper in order to claim a true breakthrough.

Response: Thanks for the reviewer's comment.

- 1) Firstly, we chose 12 mol.% CeO₂-ZrO₂ as the reinforcement/shape memory ceramic in this work. For this cerium doping concentration, the shape memory ceramic is metastable in the tetragonal phase, and is in the shape memory regime [or more precisely, the “intermediate” regime (**Fig. R12b**), rather than the superelasticity regime]. That is to say, the particles would transform to the monoclinic phase when the applied stress reaches a critical value, but would not transform back to the tetragonal phase as the applied stress is removed (**Fig. R12**).² Therefore, the tetragonal phase is thermally (and cryogenically) stable, so phase transformation cannot be triggered by thermal treatment only. Following the reviewer's suggestion, such a phenomenon was also verified via multiple thermal-cryo cycles: the increment of dislocation density during the cyclic treatment inhibits the martensitic transformation, as we described in our previous responses to Reviewer 2, Comment #8. In other words, the specific composite developed in this study is suitable for stress-induced transformations and temperature-induced reverse transformations.
- 2) Following the analysis of 1), we also carried out the mechanical loading-thermal annealing-mechanical loading stimuli, as depicted in our previous response to Reviewer 1, Question #3, where we confirmed the reversibility of dislocation motion. In addition, there was no obvious cracking in all micro-pillars after cyclic thermo-mechanical treatments (**Fig. R6**).
- 3) Finally, our statement quoted by the reviewer appears in the last paragraph when we discuss the perspective of the field and compare our results with other shape memory materials. The major

point (and also impact) of the present study is to pioneer the reversible phase transformation of shape memory ceramics in a soft metal matrix. In this study, we have already shown that this is possible, and we also proposed and justified the underlying mechanisms. The “*multiple thermal cycles and multiple mechanical cycles preferably in tension or at least in bending mode*” is not the focus of the present study. In our ongoing research, we have actually achieved the phase transformation in uniaxial tensile test, please refer to our responses to Reviewer 2, Questions #9 (**Fig. R10** and **Fig. R11c**). However, these results are out of the scope of this paper and will be published as a separate study in a future paper.

- 4) Nevertheless, the results presented in this manuscript can be considered as a breakthrough because the current work conferred the first demonstration of reversible martensitic transformation of shape memory ceramics in a densified composite without causing fracture. More encouragingly, compression on the composite micro-pillars triggered up to ~70% of the particles to transform into the monoclinic phase via a stress-induced martensitic transformation, which **breaks through the transformation limit in the granular packing (~40%)²**. Moreover, the attainment of this reversible martensitic phase transformation of SMCs constrained in light metals (Al as manifested here) may tremendously extend the application arena of these smart composites to sensors, actuators, and damping devices that require light weight, high strength, and multi-functionality. Specifically, as compared with the conventional shape memory alloys such as NiTi¹³ and Cu-Al-Ni¹⁴, the CZ/Al composite developed in this work is superior in terms of much lower density (~3.22 g·cm⁻³), and wide operational range of temperature (100-500°C), with similar actuation force/magnitude (~10² MPa). Following the reviewer’s comment, these discussions have been added to the last paragraph of the revised manuscript to manifest the

perspectives of these composites.

Fig. R12 a Schematic stress-temperature diagram for shape memory ceramics. M_s , M_f , A_s , A_f are the martensite/austenite start/finish temperature, respectively. Arrows in different colors are used to illustrate the thermo-mechanical loading scheme in the shape memory, intermediate, and superelastic regimes. **b** Transformation temperatures plotted for each of the shape memory, intermediate, and superelastic regimes for CZ shape memory ceramics at the ambient temperature investigated in the work.²

13. Page 20 line 319: “powders (“as-milled CZ/Al powders”) were put into a die”. One of the most interesting results of the work is the interface between the CZ particles and the aluminum matrix which was found to be a thin amorphous layer of Al_2O_3 . The processing conditions may play an important role and all details could be important. I suppose the die material is steel considering the very high compressive stresses used for densification? Were there any coatings or foils used to prevent sticking of the composite to the die material?

Response: As the reviewer correctly points out, the amorphous layer of Al_2O_3 at the reinforcement/Al matrix may come from the oxidization of Al during the fabrication ($2Al+3[O] \rightarrow Al_2O_3$, a spontaneous

reaction thermodynamically favorable at the ambient temperature), which plays an important role in the load transfer and plastic deformation during mechanical loading, as highlighted in our previous work¹⁵. Considering the high compressive stress (~ 600 MPa) during composite fabrication, the die material used was H13 die steel (4Cr5MoSiV1, with a yield strength of ~ 1.5 GPa at room temperature). The oily graphite was sprayed on the surface of the die to lubricate and decrease the adherence during demolding. The as-milled powders were poured into the die (inner diameter of 25 mm), followed by pre-pressing with 30 tons (~ 600 MPa) under hydraulic press. The brass wrap was heated through heating controller, thus reaching setting temperature of 480°C for hot pressing. More details of the experimental apparatus are provided in our previous work¹⁶ (**Fig. R13**).

Fig. R13 *a* Picture of the glovebox apparatus with the hydraulic press, and heating mold. *b* Detailed view of the hydraulic press with the heating mold.¹⁶

14. Fig. S1 (b) In this important graph the horizontal temperature scale is not clear. Where is the vertical line corresponding to 500°C. For the determination of the A_s , A_f and M_s temperature presumably tangent constructions were used? Fig. S1 (c) also here a temperature scale would be useful.

Response: We thank the reviewer very much for his/her careful examination on the figures. As the reviewer correctly points out, the determination of the transformation temperatures was by a tangent method¹⁷. Accordingly, we have labeled the vertical area corresponding to 500°C and added the scale bar in **Fig. S1c** of the revised supplementary materials (reproduced here as **Fig. R14b**).

Fig. R14 a Diameter distribution of CZ via particle size analyzer; **b** Phase composition of CZ extracted from the monoclinic and tetragonal characteristic peak in **c** in-situ high-temperature XRD pattern.

Reference

1. Erb DJ, Rauch HA, Knight KP, Yu HZ. Viewpoint: Tuning the Martensitic Transformation Mode in Shape Memory Ceramics via Mesostructure and Microstructure Design. *Shape Memory and Superelasticity*, (2023).
2. Yu HZ, Hassani-Gangaraj M, Du Z, Gan CL, Schuh CA. Granular shape memory ceramic packings. *Acta Materialia* **132**, 455-466 (2017).
3. Rauch HA, Chen Y, An K, Yu HZ. In situ investigation of stress-induced martensitic transformation in granular shape memory ceramic packings. *Acta Materialia* **168**, 362-375 (2019).
4. Reveron H, *et al.* Towards long lasting zirconia-based composites for dental implants: Transformation induced plasticity and its consequence on ceramic reliability. *Acta Biomaterialia* **48**, 423-432 (2017).
5. Moya X, Mathur ND. Caloric materials for cooling and heating. *Science* **370**, 797-803 (2020).
6. Lai A, Du Z, Gan CL, Schuh CA. Shape memory and superelastic ceramics at small scales. *Science* **341**, 1505-1508 (2013).
7. Du Z, Yu H, Schuh CA, Gan CL. Shape memory ceramic particles and structures formed thereof. US,SG Patent US10696599B2 (2020).
8. Williamson GK, Smallman RE. III. Dislocation densities in some annealed and cold-worked metals from measurements on the X-ray debye-scherrer spectrum. *The Philosophical Magazine: A Journal of Theoretical Experimental and Applied Physics* **1**, 34-46 (1956).
9. Du ZH, *et al.* Size effects and shape memory properties in ZrO₂ ceramic micro- and nano-pillars. *Scripta Materialia* **101**, 40-43 (2015).
10. Du ZH, Zeng XM, Liu Q, Schuh CA, Gan CL. Superelasticity in micro-scale shape memory ceramic particles. *Acta Materialia* **123**, 255-263 (2017).
11. Zeng XM, Du ZH, Tamura N, Liu Q, Schuh CA, Gan CL. In-situ studies on martensitic transformation and high-temperature shape memory in small volume zirconia. *Acta Materialia* **134**, 257-266 (2017).
12. Cho J, *et al.* In-situ high temperature micromechanical testing of ultrafine grained yttria-stabilized zirconia processed by spark plasma sintering. *Acta Materialia* **155**, 128-137 (2018).
13. Otsuka K, Ren X. Physical metallurgy of Ti-Ni-based shape memory alloys. *Progress in Materials Science* **50**, 511-678 (2005).

14. San Juan J, No ML, Schuh CA. Nanoscale shape-memory alloys for ultrahigh mechanical damping. *Nat Nanotechnol* **4**, 415-419 (2009).
15. Li Z, *et al.* Enhanced Mechanical Properties of Graphene (Reduced Graphene Oxide)/Aluminum Composites with a Bioinspired Nanolaminated Structure. *Nano Letters* **15**, 8077-8083 (2015).
16. Vogel T, Ma S, Liu Y, Guo Q, Zhang D. Impact of alumina content and morphology on the mechanical properties of bulk nanolaminated Al₂O₃-Al composites. *Composites Communications* **22**, 100462 (2020).
17. Dang P, *et al.* Low-fatigue and large room-temperature elastocaloric effect in a bulk Ti_{49.2}Ni_{40.8}Cu₁₀ alloy. *Acta Materialia*, 117802 (2022).

REVIEWER COMMENTS

Reviewer #1 (Remarks to the Author):

Review of NCOMMS-22-46469A-Z: Realizing reversible phase transformation of shape memory ceramics constrained in aluminum, by Zheng, W. et al.

The authors have responded to my previous comments in a satisfactory manner. The paper is ALMOST ready for publication. The new results have brought into focus several new issues that need to be addressed. As revisions go, these are MINOR, but important for clarity.

(1) While the authors have demonstrated their main point – reversible transformation in CZ without cracking, this has come at a cost – plastic (irreversible) deformation of the matrix (AI). Thus, any application requiring cyclic loading can only be feasible well below the strength of the pillar. This needs to be mentioned in the Discussion section.

(2) The dependence of strength on microstructure is evident (e.g., low strength of S2), but needs clear emphasis. I SUGGEST:

- Moving Table S3 (or at least its upper half) into the main text.

- Addressing the question – how can we control the microstructure? – in the Discussion section (one paragraph) and mentioning this dependence in the Abstract

(3) The overall pillar deformation upon heat treatment is not clearly stated. We have reversible phase transformation in CZ. What happens in AI? Obviously, dislocations anneal, but is any of the permanent deformation (Fig. 3b) of the pillar reversed?

(4) Text inside Fig. 1 is not visible at 100% magnification. All text needs to be larger, but the print on the shaded background is particularly bad. Also, authors should check other figures for text quality.

Reviewer #3 (Remarks to the Author):

The authors have presented some interesting work on shape memory ceramic-based composites, demonstrating thermally induced martensitic transformation on the bulk level and stress-induced transformation in micropillars. The reviewer believes that the authors have addressed the comments from previous reviewers and therefore recommends the publication of this work after addressing several minor points:

Abstract, line 24: "Our study conferred the first demonstration of reversible martensitic transformation of zirconia in a densified composite without causing fracture." I would suggest that the authors modify this sentence to clarify that the reversible stress-induced transformation is observed at a small scale, rather than the bulk level. It is less challenging to achieve martensitic transformation at a small scale without causing fracture. This clarification is especially critical when claiming to be the 'first' to achieve

such a result.

Results, line 66: "After ball milling for 3 hr, the Al powders were reshaped into thin flakes, which caused the CZ particle clusters to locate onto the surface of Al flakes." Please rephrase the sentence for clarity.

Lines 156-166: Cooling to liquid nitrogen temperature does not trigger martensitic transformation in the bulk composite. Rauch et al. claimed that cooling to liquid nitrogen temperature (soaking for 1 hr) did lead to martensitic transformation in granular packings of CZ (Journal of Applied Physics Vol. 128, 245102, 2020). Is their observation contradictory to your findings, or does it actually support your explanation that mechanical constraint is the key?

Line 185: The three pillars, S1-S3, have significantly different CZ volume fractions: 30% vs. 67% and 70%. Does this indicate a non-uniform distribution of CZ particles in the Al matrix? Why are these numbers higher than the calculated bulk level volume fraction of 16%? The authors stated that the CZ volume fraction in the pillars is determined by the STEM images in Fig. S8. It is important to consider that the CZ particles' dimension in the out-of-plane direction is smaller than the pillar's dimension, which means the CZ volume fraction should be smaller than the area fraction in the TEM image. From the X-ray tomography measurement, what is the CZ volume fraction in the bulk composite? Does it align with the calculated value (i.e., 16%)?

What potential applications do CZ composites hold for the future? It seems that the field of shape memory ceramics has been struggling to identify suitable applications to promote its growth.

Reviewer #1: The authors have responded to my previous comments in a satisfactory manner. The paper is ALMOST ready for publication. The new results have brought into focus several new issues that need to be addressed. As revisions go, these are MINOR, but important for clarity.

Response: We thank the reviewer for his/her positive comments and suggestions regarding our study. In the following, we have done our best to fully address and clarify the new issues raised. Please refer to our detailed responses as follows.

(1) While the authors have demonstrated their main point – reversible transformation in CZ without cracking, this has come at a cost – plastic (irreversible) deformation of the matrix (Al). Thus, any application requiring cyclic loading can only be feasible well below the strength of the pillar. This needs to be mentioned in the Discussion section.

Response: We are grateful to the reviewer for pointing this out, and fully agree that the reversible transformation as such comes at a cost of plastic (irreversible) deformation in the matrix. Herein, we have added a new paragraph at the end of the *Discussion* section in the revised manuscript, to identify the challenges and future opportunities for this novel class of composites. Specifically, we propose that, “*the future trends may lie in the design of the matrix materials with high intrinsic mechanical strength and large elastic strain*”. Moreover, in this paragraph, we also address the microstructural inhomogeneity of the composite in this study, and propose potential routes to improve the uniformity of the microstructure (Reviewer #1 Comment #2 in the following). The added paragraph is copied as follows:

It is noteworthy that the reversible transformation as such comes at a cost of plastic (irreversible)

deformation in the matrix. Herein, any application requiring cyclic loading can only be feasible well below the strength of the matrix. The future research direction may lie in the use of matrix materials with high intrinsic mechanical strength and large elastic strain¹. Furthermore, the CZ-Al composite used as a model material in this study was characterized by a moderate microstructural inhomogeneity, where the “force-chain” configuration of the CZ particles was lacking in some local regions in the composite, as evidenced by the pillar S2 (**Fig. S14a**) and its notably low strength than other pillars examined (**Table 1**). Undoubtedly, for more efficient phase transformation of the shape memory ceramics in the composite, the force-chain network of the ceramics should be uniformly populated across the entire body of the composite. This could be realized by fabricating ceramic preforms using vibration-assisted particle self-assembly², direct foaming³ or sacrificial templating method^{4, 5, 6} to create 3D interconnected particle networks, followed by metal infiltration to obtain an interpenetrating architecture of the two phases (i.e., interpenetrating composites, or IPCs^{7, 8}). The composite with high spatial particle connectivity is a promising candidate to realize the reversible phase transformation at the bulk level.

(2) The dependence of strength on microstructure is evident (e.g., low strength of S2), but needs clear emphasis. I SUGGEST:

- Moving Table S3 (or at least its upper half) into the main text.

- Addressing the question – how can we control the microstructure? – in the Discussion section (one paragraph) and mentioning this dependence in the Abstract

Response: We thank the reviewer for the insightful suggestions, and fully agree that the dependence of strength on microstructure needs clear emphasis in the main text. Herein, we targeted several

important aspects in **Table S3** and put them into the revised manuscript (**Table 1**, reproduced here as **Table R1**).

Table R1 Summary of CZ particle content and distribution in the pillar, and the transformational behaviors of micro-pillars S1 to S4. The actual CZ content in each pillar was estimated from the corresponding cross-section STEM images. The estimated CZ volume fractions in the micro-pillars were likely to be their upper limit, as the particles' dimension in the out-of-plane direction is smaller than the pillar's dimension.

Micro-pillar	S1	S2	S3	S4
CZ Distribution	Arc	interconnected clustering	Double arc	Laminated
Force-chain?	Yes	Yes	Yes	No
CZ Content/vol. %	30	67	70	55
Deformation mode	Load transfer through CZ chains local extrusion of Al			Al shear
Reversible transformation?	Yes	Yes	Yes	Yes

As the reviewer correctly points out, the CZ-Al composite used as a model material in this study was characterized by a moderate microstructural inhomogeneity, where the “force-chain” configuration of the CZ particles was lacking in some local regions in the composite, as evidenced by the pillar S2 (**Fig. S14a**) and its notably low strength than other pillars examined (**Table R1**). Undoubtedly, for more efficient phase transformation of the shape memory ceramics in the composite,

the force-chain network of the ceramics should be uniformly populated across the entire body of the composite. This could be realized by fabricating ceramic preforms using vibration-assisted particle self-assembly², direct foaming³ or sacrificial templating method^{4, 5, 6} to create 3D interconnected particle networks, followed by metal infiltration to obtain an interpenetrating architecture of the two phases (i.e., interpenetrating composites, or IPCs^{7, 8}).

Following the reviewer's suggestion, we have added one paragraph in the *Discussion* section to address the microstructural homogeneity (please refer to Reviewer #1 Comment #1 above for details). In addition, we have also mentioned this in the revised *Abstract* ["... (the reversible martensitic transformation of zirconia) can be interpreted by the strong geometric confinement offered by the Al matrix, the robust CZ/Al interface **and the local three-dimensional particle network/force-chain configuration that effectively transferred mechanical loads**"].

(3) The overall pillar deformation upon heat treatment is not clearly stated. We have reversible phase transformation in CZ. What happens in Al? Obviously, dislocations anneal, but is any of the permanent deformation (Fig. 4b) of the pillar reversed?

Response: The reviewer precisely noted that the dislocations in the Al matrix were annealed upon heat treatment. Indeed, as described in the previous response letter for the first round of revision, we confirmed that the dislocation generation-and-annihilation was reversible during the loading-annealing-reloading cycles (reproduced here as **Fig. R1**). Specifically, as shown, C1 was compressed only once without post-compression annealing. C2 was compressed and then annealed at 400°C. C3 went through a compression-400°C annealing-compression cycle. As manifested, these observations undoubtedly confirm the reversibility of the dislocation motion. However, the auxiliary dashed lines

in **Fig. R1a** indicate that there was barely any permanent deformation in the height of the pillars reversed (C1-C3).

As the reviewer points out in his/her Comment #1 above, the plastic (irreversible) deformation of the matrix (Al) indeed posed a serious challenge to the future application of the CZ-Al composites. We propose here that the future research direction may lie in the design of the matrix materials with high intrinsic mechanical strength and large elastic strain. This actually represents our ongoing work and is out of the scope of the present study.

Fig. R1 *a* Representative SEM image of micro-pillars C1-C3 under stress-thermal cycles. *b* TEM analysis on the low-magnification morphology, dislocations of the post-mortem micro-pillars in *a*. *c* Geometric phase analysis (GPA) analysis on micro-pillar C3, revealing the distortion of the lattice.

(4) Text inside Fig. 1 is not visible at 100% magnification. All text needs to be larger, but the print on the shaded background is particularly bad. Also, authors should check other figures for text quality.

Response: We thank the reviewer very much for his/her careful examination of the figures. We have increased the font size and enhanced the figure's contrast (reproduced here as **Fig. R2**). In the final version of the manuscript, we will also make sure to upload the original, high-definition images with at least 300 dpi resolution.

Fig. R2 Formation, 3D architecture, and phase constitute of the as-fabricated CZ/Al composites. *a* Schematic illustration of the as-synthesized shape memory Ce-doped zirconia (CZ) particles, the as-milled CZ/Al mixed powders, and the as-fabricated CZ/Al composite. The representative scanning electron microscopy (SEM) images for each stage during fabrication are also provided in **b**; **c** High-resolution X-ray computed tomography microscopy (XRM) volume renderings of CZ/Al composites. **d**

Electron back scattering diffraction (EBSD) of CZ/Al and pure Al, mapped with Kikuchi band contrast and crystal orientation contrast (Z axis). The grain size of Al was fitted and shown in e; f phase constitution at room temperature (25°C) measured by X-ray diffraction (XRD)

Reviewer #3: The authors have presented some interesting work on shape memory ceramic-based composites, demonstrating thermally induced martensitic transformation on the bulk level and stress-induced transformation in micropillars. The reviewer believes that the authors have addressed the comments from previous reviewers and therefore recommends the publication of this work after addressing several minor points:

Response: We thank the reviewer very much for his/her positive evaluation on our manuscript, and have done our best to address the reviewer's questions/comments in the following.

1. Abstract, line 24: "Our study conferred the first demonstration of reversible martensitic transformation of zirconia in a densified composite without causing fracture." I would suggest that the authors modify this sentence to clarify that the reversible stress-induced transformation is observed at a small scale, rather than the bulk level. It is less challenging to achieve martensitic transformation at a small scale without causing fracture. This clarification is especially critical when claiming to be the 'first' to achieve such a result.

Response: We greatly appreciate and strongly endorse the reviewer's suggestions. In the revised manuscript, we modified all the related sentences to emphasize that the new finding of reversible martensitic transformation was observed at a small scale. In particular, we have revised the sentence in the *Abstract* to be "Our study conferred the first demonstration of reversible martensitic transformation of zirconia **in densified composite micro-pillars** without causing fracture, which can be interpreted by the strong geometric confinement offered by the Al matrix, the robust CZ/Al interface and the local three-dimensional particle network/force-chain configuration that effectively transferred mechanical loads, and the decent flowability of the matrix that accommodated the volume change

during phase transformation.”

In addition, perspectives on realizing reversible martensitic transformation at the bulk level have also been added to the *Discussion* part. Thank you.

2. Results, line 66: "After ball milling for 3 hr, the Al powders were reshaped into thin flakes, which caused the CZ particle clusters to locate onto the surface of Al flakes." Please rephrase the sentence for clarity.

Response: We thank the reviewer for his/her careful examination on the language and readability of our manuscript. We have corrected the sentence in the revised manuscript and reproduced it here: “*After ball milling for 3 hr, the Al powders were reshaped into thin flakes, and the CZ particle clusters were embedded on the surface of these Al flakes*”. Following the reviewer’s comment, we have also done a thorough inspection on the language of the manuscript to improve its quality.

3. Lines 156-166: Cooling to liquid nitrogen temperature does not trigger martensitic transformation in the bulk composite. Rauch et al. claimed that cooling to liquid nitrogen temperature (soaking for 1 hr) did lead to martensitic transformation in granular packings of CZ (Journal of Applied Physics Vol. 128, 245102, 2020). Is their observation contradictory to your findings, or does it actually support your explanation that mechanical constraint is the key?

Response: We thank the reviewer very much for pointing this out. In the forward-looking work raised by Rauch et al.⁹, thermally induced martensitic transformation was evident in both granular packings made of polycrystalline particles (particle size $\sim 100\ \mu\text{m}$ or above) or single crystal particles (1-5 μm). On one hand, the CZ particles in this work have smaller size ($\sim 0.5\ \mu\text{m}$), rendering a lower martensite

transformation temperature. On the other hand (also more importantly), as the reviewer points out, we believe that matrix (mechanical) constraint was the key for the stabilization of the tetragonal phase in our composite. Specifically, the ceramic grains in the granular packing suffer from debonding under cyclic stimuli, while the sandwiched interface (CZ/Al₂O₃/Al) in the bulk composites in this work endowed the composite with strong bonding state to stabilize the tetragonal structure, effectively slashing the martensite transformation temperature.

For the reviewer's information, in our ongoing research, we have actually quantified the matrix constraint from Al and other metals by applying different cold-pressing stress, sintering temperature, cerium dopant ratio and CZ weight ratio in the composites. Indeed, the matrix constraint as such has been proved to contribute to the austenitization and stabilization of austenite. The results will be published as a separate paper in the future.

Following the reviewer's comments, we have added the work raised by Rauch et al. into the *Discussion* part to better claim the difference. The added sentence is copied here: "*Such a geometric constraint successfully inhibited the $t \rightarrow m$ phase transformation (Fig. S15a, b), exhibiting a totally different phenomenon from that of granular packings⁹.*"

4. Line 185: The three pillars, S1-S3, have significantly different CZ volume fractions: 30% vs. 67% and 70%. Does this indicate a non-uniform distribution of CZ particles in the Al matrix? Why are these numbers higher than the calculated bulk level volume fraction of 16%? The authors stated that the CZ volume fraction in the pillars is determined by the STEM images in Fig. S8. It is important to consider that the CZ particles' dimension in the out-of-plane direction is smaller than the pillar's dimension, which means the CZ volume fraction should be smaller than the area fraction in the TEM image. From

the X-ray tomography measurement, what is the CZ volume fraction in the bulk composite? Does it align with the calculated value (i.e., 16%)?

Response: We thank the reviewer very much for his/her reminder and advice. In essence, the non-uniform distribution of CZ particles in the Al matrix stemmed from the specific processing route, i.e., the low-speed ball milling followed by densification. In the resulting composite, the CZ particles were preferably located at the Al-Al grain boundaries, forming a network structure, as demonstrated at a large view in **Fig. S3** (reproduced here as **Fig. R3**). As we mentioned in the main manuscript, “The stronger micro-pillars S1-S3 as compared to the corresponding bulk composite may be caused by the effectively higher CZ content in the pillar. In particular, the regions rich in CZ particle clusters (as depicted in **Fig. 1c**) were generally selected for micro-pillar fabrication”. In other words, in this study, we intentionally selected regions rich in CZ particles for micro-pillar fabrication, for better demonstration of the reversible phase transformation.

As both Reviewer #1 and Reviewer #3 point out, the composite fabricated in this study was characterized by a moderate microstructural inhomogeneity, where the “force-chain” configuration of the CZ particles was lacking in some local regions in the composite. As we respond to Reviewer #1 Comment #2, for more efficient phase transformation of the shape memory ceramics in the composite, the force-chain network of the ceramics should be uniformly populated across the entire body of the composite. Following the reviewers’ comments and suggestions, we have added a separate paragraph in the revised *Discussion* section, to address the microstructural inhomogeneity and propose the potential routes to improve the uniformity of the microstructure (please refer to Reviewer #1 Comment #1 and Comment #2 above for details).

Fig. R3 *a* SEM image of CZ/Al with a large view, showing the typical particle network structure at 2D plane; *b* Electron back scattering diffraction (EBSD) of CZ/Al in a magnified region with *c* corresponding elemental mapping by energy dispersive spectroscopy (EDS) in SEM, indicating that the CZ particles were preferably dispersed along the Al grain boundary.

As the reviewer correctly points out, the CZ particles' dimension in the out-of-plane direction is smaller than the pillar's dimension, indicating a smaller CZ volume fraction than the area fraction estimated from the STEM image. Therefore, following the reviewer's suggestion, we have added a couple of sentences in the revised manuscript, "The actual CZ content in each pillar was estimated from the corresponding cross-section STEM images. The estimated CZ volume fractions in the micro-pillars were likely to be their upper limit, as the particles' dimension in the out-of-plane direction is smaller than the pillar's dimension" (captions of revised **Table 1**, reproduced as **Table R1** in this letter). In addition, from X-ray tomography, the CZ volume fraction was estimated to be ~15.4%, aligning

well with the prescribed value from composite fabrication. We have added this in the revised manuscript when talking about the X-ray tomography results (**Fig. 1** figure caption).

5. What potential applications do CZ composites hold for the future? It seems that the field of shape memory ceramics has been struggling to identify suitable applications to promote its growth.

Response: We thank the reviewer very much for his/her constructive questions. As the reviewer insightfully pointed out, the field of shape memory ceramics has been struggling to identify suitable applications to promote its growth. In particular, over the past decades, researchers have been dedicated to overcoming two intrinsic drawbacks of shape memory ceramics, i.e., to mitigate their brittleness via the improvement of specific surface area and/or the removal of grain boundaries^{10, 11}, and the other is to lessen the transformation hysteresis via the refinement of kinematic/geometric compatibility between neighboring phases/grains^{12, 13}).

Regarding potential applications of shape memory ceramics, from the perspective of smart materials, the compositing method enlighten the pathway for shape-memory effect and superelasticity at a large scale, which may extend the application arena to sensors, actuators, and damping devices that require lightweight, high strength, and multi-functionality^{14, 15}. From the perspective of structural applications, the transformation-induced large lattice strains of shape memory ceramics (~10% in shear strain and ~5% in volume change¹³) may render the derived metal-, ceramic-, or polymer-matrix composites with high strength-toughness synergy via transformation toughening¹⁶. These thoughts and considerations have been included in the conclusion of the revised manuscript, following the reviewer's comment.

Reference:

1. Hao S, *et al.* A Transforming Metal Nanocomposite with Large Elastic Strain, Low Modulus, and High Strength. *Science* **339**, 1191-1194 (2013).
2. Muto H, Kimata K, Murata K, Daiko Y, Matsuda A, Sakai M. Fabrication of three-dimensionally close-packed aggregate of particles under mechanical vibration. *Materials Science and Engineering: B* **161**, 193-197 (2009).
3. Rauch HA, *et al.* Additive manufacturing of yttrium-stabilized tetragonal zirconia: Progressive wall collapse, martensitic transformation, and energy dissipation in micro-honeycombs. *Additive Manufacturing* **52**, 102692 (2022).
4. Zhao X, Lai A, Schuh CA. Shape memory zirconia foams through ice templating. *Scripta Materialia* **135**, 50-53 (2017).
5. Chen S, *et al.* Status and strategies for fabricating flexible oxide ceramic micro-nanofiber materials. *Materials Today* **61**, 139-168 (2022).
6. van Bommel KJC, Friggeri A, Shinkai S. Organic Templates for the Generation of Inorganic Materials. *Angewandte Chemie International Edition* **42**, 980-999 (2003).
7. Bauer J, Sala-Casanovas M, Amiri M, Valdevit L. Nanoarchitected metal/ceramic interpenetrating phase composites. *Science Advances* **8**, eabo3080 (2022).
8. Zhang M, *et al.* 3D printed Mg-NiTi interpenetrating-phase composites with high strength, damping capacity, and energy absorption efficiency. *Science Advances* **6**, eaba5581 (2020).
9. Rauch HA, Yu HZ. Effects of mechanical constraint on thermally induced reverse martensitic transformation in granular shape memory ceramic packings. *Journal of Applied Physics* **128**, 245102 (2020).
10. Lai A, Du Z, Gan CL, Schuh CA. Shape memory and superelastic ceramics at small scales. *Science* **341**, 1505-1508 (2013).
11. Erb DJ, Rauch HA, Knight KP, Yu HZ. Viewpoint: Tuning the Martensitic Transformation Mode in Shape Memory Ceramics via Mesostructure and Microstructure Design. *Shape Memory and Superelasticity*, (2023).
12. Gu H, *et al.* Exploding and weeping ceramics. *Nature* **599**, 416-420 (2021).
13. Pang EL, Olson GB, Schuh CA. Low-hysteresis shape-memory ceramics designed by multimode modelling. *Nature* **610**, 491-495 (2022).

14. Yu HZ, Hassani-Gangaraj M, Du Z, Gan CL, Schuh CA. Granular shape memory ceramic packings. *Acta Materialia* **132**, 455-466 (2017).
15. Du Z, Yu H, Schuh CA, Gan CL. Shape memory ceramic particles and structures formed thereof. US,SG Patent US10696599B2 (2020).
16. Kelly PM, Rose LRF. The martensitic transformation in ceramics - its role in transformation toughening. *Progress in Materials Science* **47**, 463-557 (2002).

REVIEWERS' COMMENTS

Reviewer #1 (Remarks to the Author):

The authors have addressed my comments adequately. I recommend that the paper be accepted.

Reviewer #3 (Remarks to the Author):

The authors have addressed the reviewer's previous comments. The manuscript is ready for publication.